# Landslide hazard early warning method for rock slopes using a hybrid LSTM-SARIMA data-driven model

Yongxin Dai[1,2,3,4], Zijian Li[1,2,3]*, Jingbiao Lu[1,2,3]

1 Sinosteel Maanshan Mining Research Institute Co., Ltd., Maanshan, China, 2 National Key Laboratory of Metal Mine Safety and Disaster Prevention and Control, Maanshan, Anhui, China, 3 Huawei National Engineering Research Center for Efficient Recycling of Metal Mineral Resources Co., Anhui Sheng, China, 4 College of Civil Engineering and Hydraulic Engineering, Shandong University, Jinan, China

* 994693520@qq.com

## Abstract

Rock slope landslides are characterized by their sudden onset and significant destructive power, posing a major threat to human life as well as the safety of equipment and infrastructure.Currently, research on landslide early hazard warning has largely focused on individual components, such as monitoring data analysis or studies on influencing mechanisms. However, landslide early hazard warning is a complex, multi-stage technical system where each stage is closely interlinked, and focusing solely on a single component cannot fulfill the objectives of effective monitoring and warning. This paper proposes a comprehensive technical system for landslide early hazard warning in open-pit mine slopes, encompassing the full process of monitoring data acquisition and processing, analysis of influencing mechanisms, intelligent algorithm-based prediction, and the construction of early hazard warning indicators. Each stage of the early hazard warning process is systematically researched and summarized.First, the combination of sliding average and wavelet noise reduction is utilized to perform global denoising and local focus noise reduction on the original monitoring data, and the signal-to-noise ratios after two rounds of noise reduction are 36 and 44, respectively, which indicates a good noise reduction effect. The Hodrick-Prescott (HP) filter is used to split the slope displacements into components, the Long Short-Term Memory (LSTM)–Seasonal Autoregressive Integrated Moving Average (SARIMA) hybrid model is proposed to predict the slope of the trend term of displacements and period term of displacements, and the prediction accuracy of the LSTM–SARIMA hybrid model reaches 96%. The excellence of the hybrid-driven model was determined by introducing five data-driven models, a Support Vector Machine (SVM), a Random Forest (RF),eXtreme Gradient Boosting (XGBoost),Recurrent Neural Network(RNN) and Light Gradient Boosting Machine(LightGBM), for comparison.Finally, the improved tangent angle of the T-t curve is employed as the landslide warning criterion, enabling accurate prediction of

**Data availability statement:** All relevant data are within the article and its Supporting Information files.

**Funding:** The author(s) received no specific funding for this work.

**Competing interests:** The authors have declared that no competing interests exist.

landslide events in an open-pit mine in East China. The successful application of this system demonstrates that the comprehensive warning framework proposed in this study can accurately predict the occurrence of rock slope landslides.

## 1. Introduction

Mineral resources are important pillars for the survival and development of human society and guarantee national economic and social development. With the development of science and technology, mining is moving toward increasingly deeper levels, accompanied by the frequent occurrence of geological disasters [1]. Landslide disasters are among the most common geological disasters in the production process of mines, and the in-depth development of open-pit mine slope slippage early hazard warning technology research to promote the efficient mining of mineral resources and reduce the loss of people and property in open-pit mining has important theoretical and practical significance [2–4].

Landslide disaster early hazard warning is a complex technical system that encompasses data preprocessing, including noise reduction and data cleaning, analysis of deformation influencing factors, selection of intelligent prediction algorithms, and the development of early hazard warning guidelines. Each of these technologies represents a critical component within the landslide early hazard warning system, with each element being interconnected and mutually supportive.The application of early hazard warning technologies for landslide hazards enables the timely detection of precursory signs of landslide occurrence,and countermeasures can be taken in advance by evaluating the danger of slopes, which is currently considered to be the most effective technique for reducing the hazard of landslides [5–7].

Since the 1980s, with the development of modern mathematical theory, mathematical statistics, probability theory, fuzzy mathematics, gray correlation methods, etc., have been widely used in landslide prediction, and scholars worldwide have made inferences based on mathematical theories and established several landslide prediction models [8].In recent years,He Keqiang et al. [9] developed a two-parameter statistical prediction model based on slope displacement vector angle and displacement rate, employing principles of mathematical and statistical trend analysis.This model addresses the limitations of traditional single-parameter prediction theories in displacement time-series analysis.Malamud et al. [10] used mathematical statistics and probabilistic models to study the spatial and temporal distributions of landslides and developed many mathematical models to describe landslide phenomena. Li Tianbin et al. [11] established the Fernhas inverse function model based on the cumulative displacement–time curve of landslides being opposite of the Fernhas curve. Landslide prediction based solely on mathematical methods often focuses on the internal logic of the monitoring data itself and is rarely connected to the underlying mechanisms of landslide deformation. Given that landslides are complex processes influenced by multiple factors, relying exclusively on data analysis may fail to accurately represent the objective conditions of landslide deformation, thereby reducing predictive accuracy.

Slope slip is a complex nonlinear problem affected by multiple factors, and many scholars have begun to use nonlinear models to predict slope displacements. Artificial intelligence-based methods with a strong ability to address nonlinear problems, such as time series regression, functional networks, artificial neural networks (ANNs) and support vector machines (SVMs).Xu et al. [12] were the first to introduce an improved neural network model, specifically an autoregressive time series model, which combines traditional neural networks with rock creep theory to predict landslide displacement. Du et al. [13] utilized a time series model to separate cumulative displacement into trend and seasonal components, primarily considering displacement due to the self-weight of rock and displacement triggered by rainfall.Krkaanov et al. [14] successfully proposed another random forest (RF)-based landslide motion prediction method. This method takes into account multiple influencing factors, including rock creep, human activities, rainfall, and groundwater.AI algorithms have been optimized in recent years. Zhang et al. [15] used a gradient boosting decision tree algorithm implemented based on the LightGBM framework.The missing data were filled in by interpolation, and the historical feature data were utilized to generate near-term features and far-term features to obtain the importance ranking of the factors affecting the slope evolution process and the optimal parameter set of the algorithm.Liang et al. [16] used synthetic aperture radar interferometry (InSAR) technology. Combined with hotspot analysis and machine learning algorithms, research on the automatic extraction of regional high-deformation slopes and artificial intelligence identification of potential landslides was conducted to improve the efficiency and accuracy of potential landslide identification.Although researchers have made some progress in predicting slope displacement by utilizing intelligent algorithms in conjunction with influencing mechanisms, displacement prediction is merely a means to an end, with early hazard warning as the ultimate objective. Landslide early hazard warning is an ongoing technical system, and without well-defined early hazard warning guidelines, predictive results hold limited significance for engineering applications.

Recent studies have attempted to bridge this gap by integrating geomechanical insights with AI-based predictions. For instance, Xiang et al. [17]analyzed the failure mechanisms of step-like landslide displacements under rainfall and reservoir water level dynamics, providing a novel approach for integrating physical insights with predictive modeling. Similarly, Xiang et al. [18] proposed a hybrid support vector regression model optimized with factor preprocessing to enhance the prediction of step-like displacements in the Three Gorges Reservoir area, emphasizing the importance of unsaturated soil mechanics in deformation analysis. Furthermore, Wen et al. [19] developed a hybrid PSO-GSA-SVR model utilizing singular spectrum analysis to improve landslide displacement forecasting, demonstrating its effectiveness in the Jiuxianping landslide case study. Lei et al. [20]proposed a method based on Bayesian ensemble learning and Shapley additive explanations for rapid slope stability estimation, and developed a user-friendly graphical interface tool to simplify the assessment process. Long et al. [21] compared the application of supervised classification methods in landslide evolution research in the Mianyu River basin, China, and found that the Random Forest algorithm performed excellently in landslide identification. Additionally, Ma et al. [22] introduced an automated machine learning (AutoML) technique to propose an easy-to-use landslide susceptibility mapping method that enhances model accuracy and user-friendliness through automation. Other studies have shown that selecting appropriate unit types significantly affects the results of landslide susceptibility mapping. Ma et al. [23] compared the application of slope units and grid units in landslide events in Gansu Province, China, and found that slope units effectively reduced model uncertainty, thereby improving prediction accuracy.These studies provide new methodologies and perspectives for landslide displacement prediction, particularly in addressing the nonlinear and complex behavior of landslides. By combining artificial intelligence algorithms with factor preprocessing techniques, these approaches have improved prediction accuracy and reliability. However, current research still faces challenges, including the lack of a unified early hazard warning standard and insufficient integration of monitoring data with deformation mechanisms. Therefore, further developments require establishing a closer connection between physical mechanism analysis and data-driven models to enhance the accuracy and practicality of landslide early hazard warning systems.

Open-pit mine landslide early hazard warning is a comprehensive technical system that integrates monitoring data processing and analysis, impact mechanism analysis, intelligent prediction, and precise warning. This process requires

multi-level integration and application of various technologies to achieve accurate landslide warnings. Current research often focuses on individual aspects: either on mathematical algorithms addressing the data itself, often lacking mechanism analysis, or on slope displacement trend prediction through intelligent algorithms combined with intrinsic mechanism considerations, which still falls short of achieving precise early hazard warnings. Addressing these limitations, this paper proposes a comprehensive approach to the landslide early hazard warning process. This involves sequentially performing data acquisition, cleaning, and noise reduction, identifying monitoring data composition based on landslide mechanism analysis, and accurately predicting slope displacement trends using a hybrid data-driven model. Finally, this paper proposes the use of an improved tangential angle as an early hazard warning criterion and establishes an intelligent workflow for landslide early hazard warning.

## 2. Methods

This paper establishes a technical system for early hazard warning of slope landslide disasters in open-pit mines, detailing the complete workflow from monitoring data acquisition and processing, impact mechanism analysis, and intelligent algorithm prediction to the construction of early hazard warning indicators, ensuring accurate prediction and assessment of potential landslides. Continuous radar displacement monitoring data from an open-pit mine in East China, collected over more than two years, serves as the basis for analysis. The first step involves data cleaning and noise reduction, employing a combination of sliding average and wavelet denoising algorithms to achieve both global and local denoising, thereby ensuring data authenticity and purity—a critical step for accurate analysis. The second step addresses the analysis of influence mechanisms, as rocky slope landslides are affected by factors such as rock self-weight, rainfall, and human activities. Here, the cleaned displacement data are decomposed into components to extract the gravitational displacement trend term and the periodic displacement term under environmental influence, allowing for differentiated analysis of the effects of various factors. The third step entails developing an intelligent prediction algorithm to forecast slope displacement over future spatial and temporal intervals. Given the diverse influences on slope displacement, its behavioral characteristics vary, necessitating a hybrid data-driven model for optimal predictive accuracy. In this study, the LSTM-SARIMA hybrid model is used to predict the displacement trend and periodic terms separately, with results weighted and summed to obtain the total displacement. To validate the hybrid model's effectiveness, its predictive accuracy is compared against SVM, XGBoost, and RF models. The final step establishes a landslide warning index, where selecting an appropriate index is critical for precise early hazard warnings to mitigate life and property risks. This study introduces an improved tangent angle as a warning indicator by transforming the predicted displacement-time curve to obtain an enhanced T-t curve and defining the tangent angle according to Saito's three-phase theory for accurate warning. By integrating each component of landslide early hazard warning, this paper develops a complete technical system and applies it to an open-pit mine in East China, successfully providing an accurate landslide disaster warning. A technical flowchart of this paper is shown in Fig 1.

### 2.1 Data preprocessing

Data collection and processing are the first and most crucial steps in digital analysis. The quality, completeness and accuracy of the data directly determine the reliability and validity of the analysis results of the data-driven model. In open pit mines, radar monitoring data acquisition is affected by falling rocks, human activities, rainfall and other factors, and there is considerable noise in the monitoring data, which is detrimental to the digital analysis process. Therefore, monitoring data must be preprocessed to remove noise and restore the most accurate state of the data before carrying out digital analysis.

Denoising is applied to radar displacement monitoring data. Given the complexity and irregularity of this data, which is influenced by various factors, a single denoising method is insufficient to fully eliminate noise from the monitoring data. In this paper, the sliding average filter–wavelet denoising combined work method and noise reduction process are

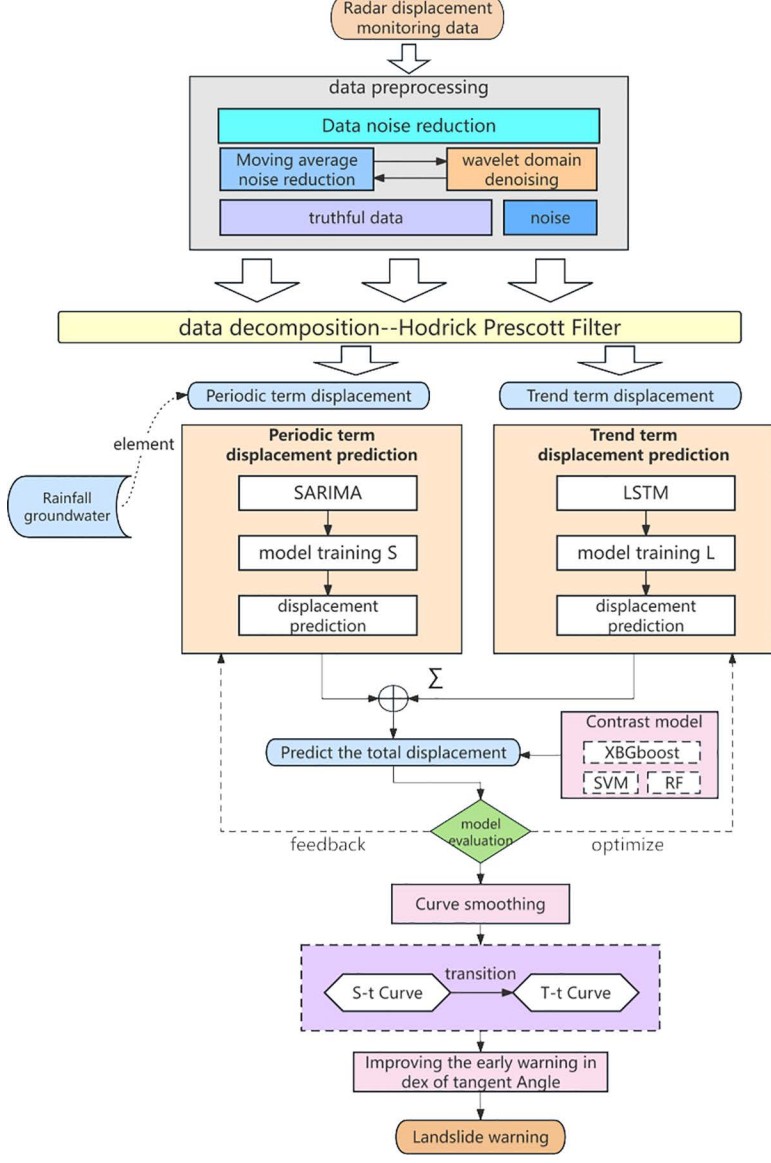

**Fig 1. Technology roadmap.**

shown in Figs 2–4. Sliding average filtering for high-frequency noise has a better inhibitory effect, can smooth the signal, and can achieve comprehensive analysis of the global signal for overall denoising. The denoising wavelet transform can perform multiscale analysis in the time-frequency domain, making it possible to better capture the local characteristics of the signal. Therefore, the sliding average filtering-wavelet denoising method can be combined with global noise reduction and local focus noise reduction to effectively denoise monitoring data and provide excellent datasets for subsequent digital analysis.It is worth noting that when combining moving average with wavelet denoising, there is an issue of parameter complexity. This complexity stems from the coupling of multiple parameters, the explosion of the search space, and conflicting objectives, which result in a reliance on empirical knowledge or the consumption of substantial resources for parameter tuning in practical applications. If computational resources are limited, alternative denoising algorithms, such as Kalman filtering or adaptive filtering, may be considered for data processing.

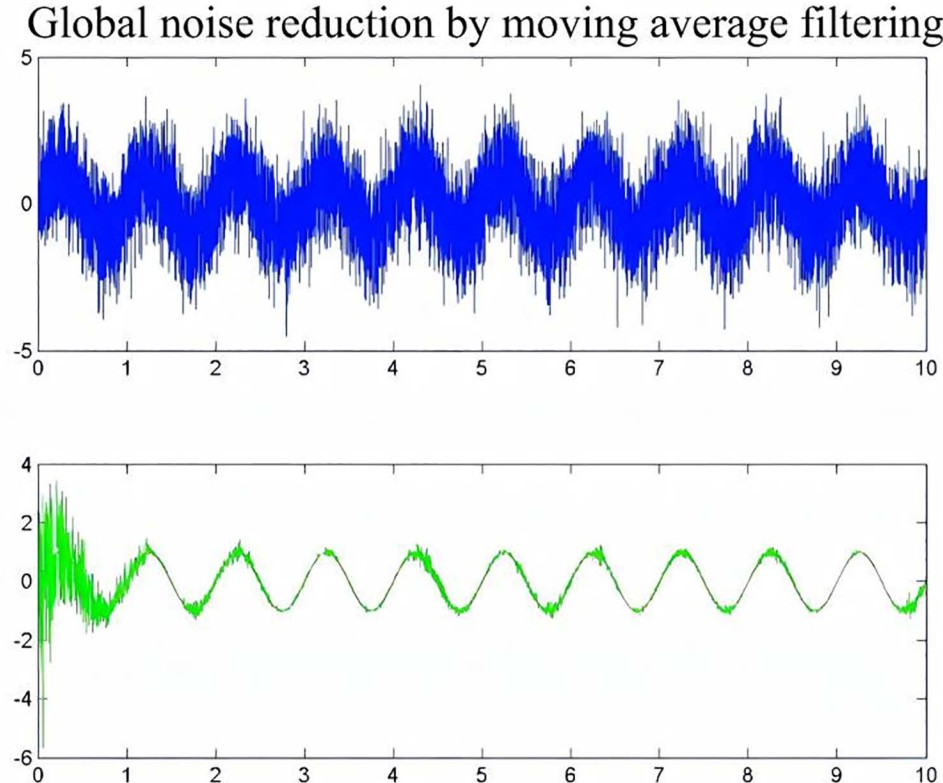

**Fig 2. Noise reduction by the sliding average algorithm.**

The monitoring data are first subjected to a global integrated denoising process, and sliding average filtering is a simple filtering method that averages the signal to remove noise. Specifically, sliding average filtering divides the signal into numerous windows, and the data within each window are averaged to obtain an average value as the output value of the window. As the window slides forward, new data are added, and old data are removed, thus realizing the real-time processing of the signal. Then, localized focused noise reduction is performed on the monitoring data using a wavelet denoising algorithm. The fundamental principle of wavelet threshold denoising involves applying a wavelet transform to the signal, which results in wavelet coefficients that capture crucial details about the signal. Upon decomposition, the signal's significant wavelet coefficients emerge, while those corresponding to noise remain smaller. Typically, the noise wavelet coefficients are less than those of the signal. By choosing an appropriate threshold, the wavelet coefficients that exceed this threshold are presumed to originate from the signal and are thus preserved. Conversely, coefficients below the threshold are assumed to stem from noise and are set to zero, effectively reducing noise in the signal.That is

$$WT_x\left(\alpha, \tau\right) = \frac{1}{\sqrt{\alpha}} \int_{-\infty}^{+\infty} x\left(t\right)\varphi\left(\frac{t-\tau}{\alpha}\right) dt$$

(1)

In the above equation, $\alpha > 0$ is the scale factor, which serves to scale the basic wavelet $\varphi\left(t\right)$ -function, reacting to the displacement, the value of which can be positive or negative. Since both variables are continuous, they are referred to as continuous wavelet transforms.

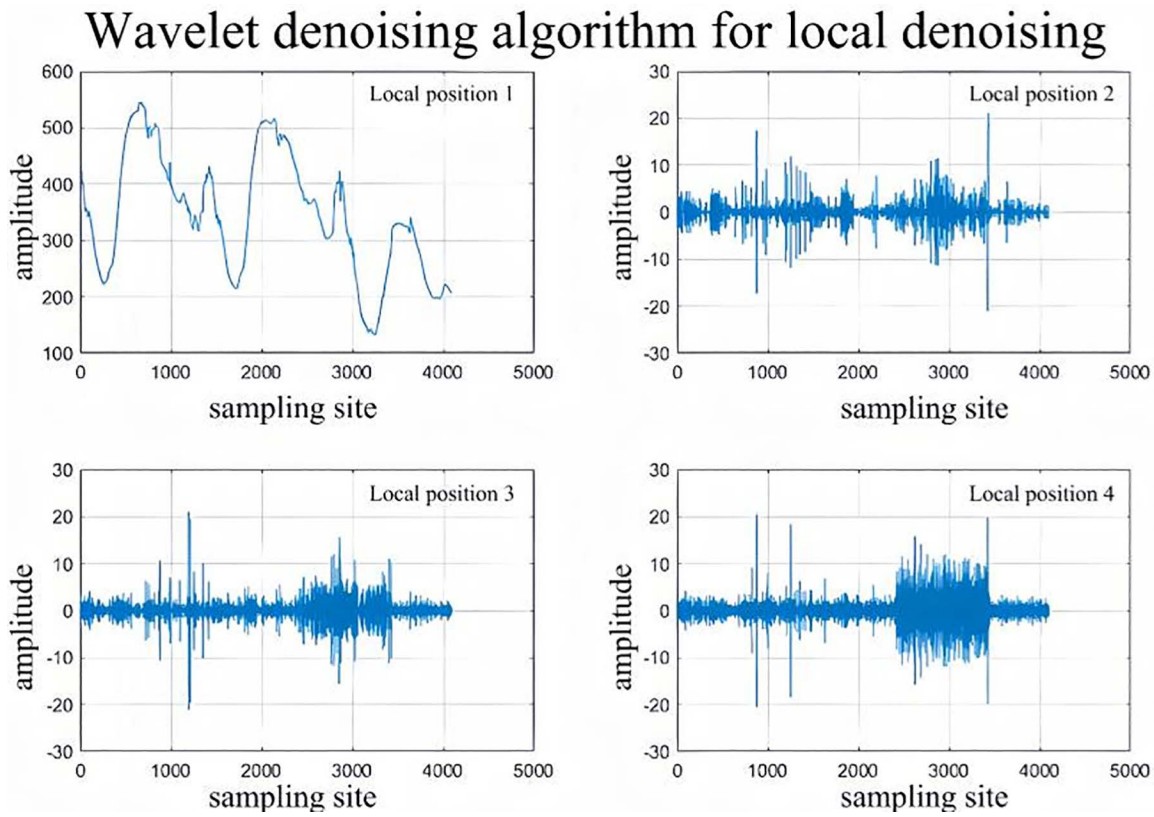

**Fig 3. Wavelet noise reduction.**

## 2.2 Slope displacement prediction

**2.2.1 Trend term–periodic term separation.** Under the joint influence of external triggering factors (rainfall, temperature, human activities, etc.), internal change factors of the slope body (seepage pressure, groundwater level, soil water content, etc.) and its own geologic conditions (topography, geological structure and stratigraphic lithology), the deformation and displacement of the landslide show uncertainty and randomness and are not easy to predict. The factors influencing landslide deformation have different modes of influence; their own geological conditions determine the overall trend of landslide deformation, and the internal and external influencing factors lead to irregular changes in landslide displacement. In the past, several scholars used a simple and rough way to accumulate all the factors affecting landslides and did not differentiate between them, which led to problems such as poor prediction effects and difficulty in prediction. Therefore, in this paper, the cumulative landslide displacement is decomposed into two parts, the trend term and the change term: the trend term of displacement is an approximate monotonic growth curve over time, and the change term of displacement is a more complex nonlinear time series.

In this paper, the trend term of displacement and the periodic term of displacement in the displacement data are separated using the HP filter, which refers to the data smoothing technique. The HP filter is a time-series analysis tool commonly used in economics to decompose raw data into a trend portion and a periodic fluctuation portion. It works by fitting a smoothed trendline by minimizing the trade-off between volatility and trend in the data. This method helps analysts better understand long-term trends in data and remove short-term fluctuations. The formula for the HP filter is given below:

$$y_t = \tau_t + c_t \qquad (2)$$

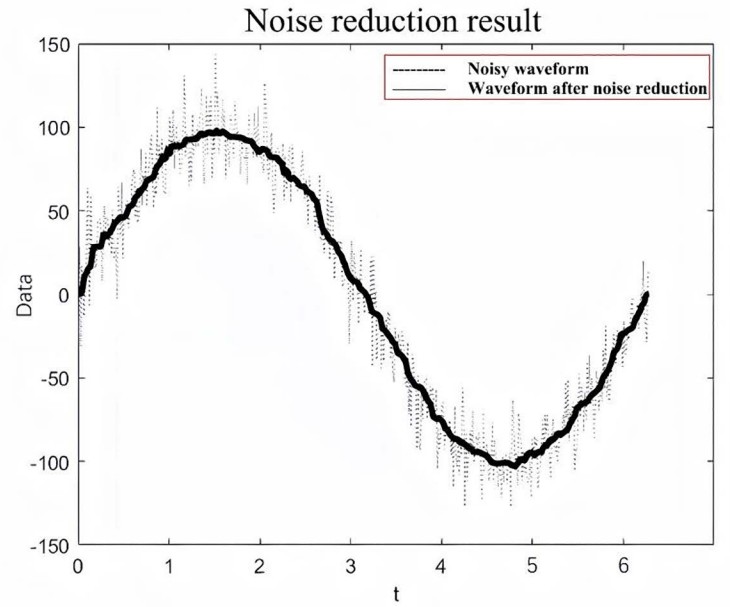

**Fig 4. Effect of noise reduction after two rounds.**

$y_t$ is the observed raw data; $\tau_t$ is the trend component, which represents the long-term trend; and $c_t$ is the periodic fluctuation component.

The optimization objective of the HP filter is to minimize the error of the following equation:

$$\min imize \sum_{t=1}^{T} (y_t - \tau_t)^2 + \lambda \sum_{t=2}^{T-1} [(\tau_{t+1} - \tau_t) - (\tau_t - \tau_{t-1})]^2 \tag{3}$$

where $\lambda$ is a smoothing parameter that weighs the smoothness of the trend against the fit of the data. A larger $\lambda$ will result in a smoother trend, and a smaller $\lambda$ will emphasize the fit of the data more. The fitting results corresponding to different values of $\lambda$ are shown in Fig 5.

The working principle of the HP filter can be roughly summarized in the following steps:

① The raw data are split into a trend portion and a periodic fluctuation portion.

② An error function is minimized by adjusting the trend part to minimize the error between the original data and the fitted value.

③ The choice of the smoothing parameter $\lambda$ is crucial to the filtering effect and is determined according to the characteristics of the data and the purpose of the analysis.

**2.2.2 Projecting the trend term of displacement.** The trend term of displacement is a regular and monotonically increasing trend generated by the gravity of the slope rock mass itself, and its displacement curve has a certain regularity. The regular change in the trend term of displacement is an important monitoring index in geoengineering that can help engineers better assess the stability of rock mass and take necessary preventive and management measures. Through long-term and systematic monitoring and analysis, the displacement trend of the rock body can be accurately predicted and controlled to ensure the safety and sustainable development of the project. Therefore, in-depth research and understanding of the trend term of the displacement law are highly important for geoengineering practice.

 

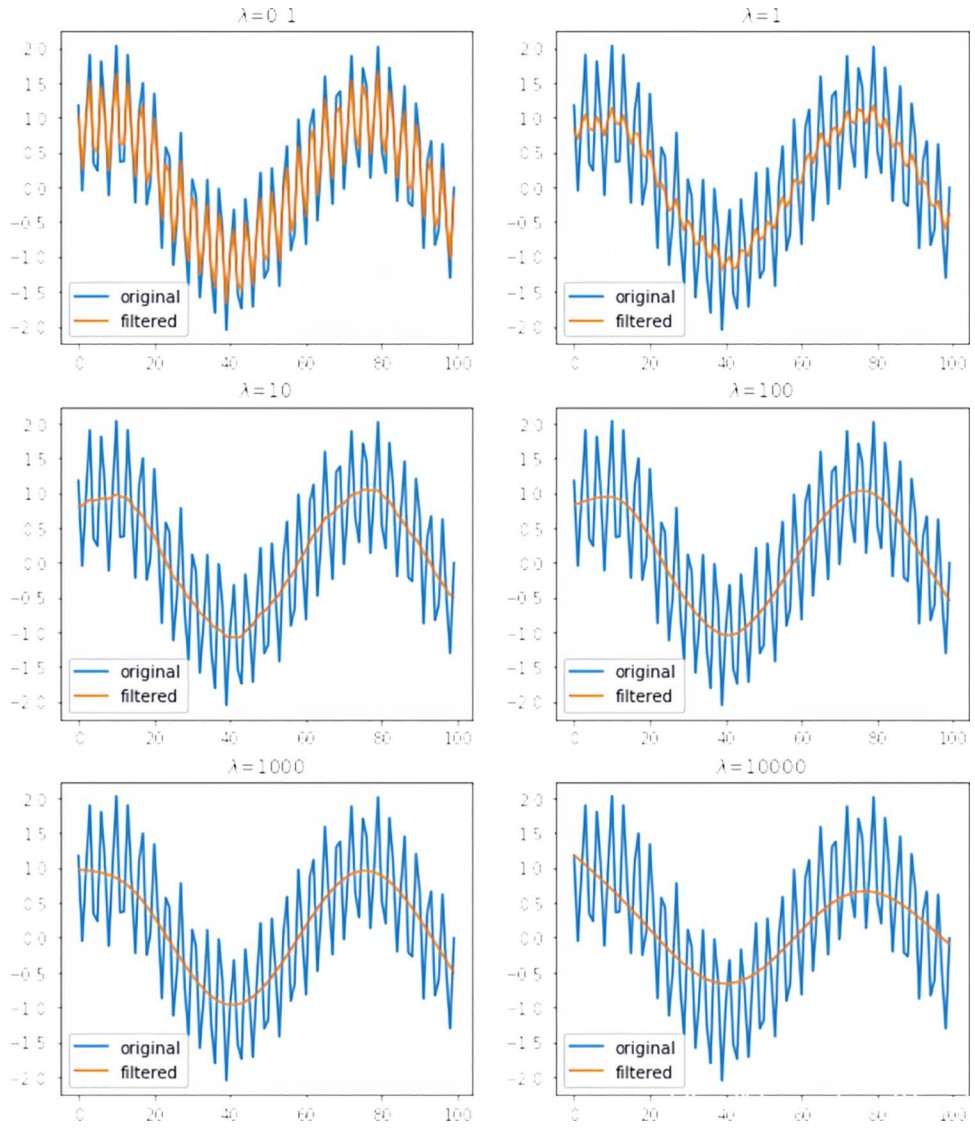

**Fig 5.** Results of curve fitting for different sizes $\lambda$.

The obvious regularity of the trend term of displacement implies that there is a long-term dependency between the displacement value at the current moment and the displacement value at the previous moment. The LSTM network, which can memorize and learn long-term dependencies, can effectively capture such dependencies and use historical data to predict future values. Moreover, monotonically increasing trends are often accompanied by complex nonlinear relationships, and LSTM, as a powerful nonlinear model, can flexibly adapt to and capture nonlinear features in the data to make more accurate predictions.

LSTM is a recurrent neural network (RNN) variant for processing sequence data and is particularly suitable for modeling and predicting long sequences. Compared to traditional RNNs, LSTM is better able to capture long-term dependencies in sequences when processing long sequences while avoiding problems such as gradient vanishing or gradient explosion [24].

The core of the LSTM network is a unit, and each unit contains three gates and a cell state, which work together to control the flow of information and memory. These three gates are the input gate, forget gate and output gate, which control the input, forgetting and output of information, respectively, through the weights obtained from learning. The state unit is then responsible for recording and transmitting the information. The structure of the LSTM network is shown in Fig 6.

The oblivion gate decides which information should be retained or discarded. It receives information from the previous hidden state and the current input and outputs a value between 0 and 1 via a sigmoid function, where 0 means completely forgotten and 1 means completely retained.

The input gate decides which information should be added to the cell state. It also receives information from the previous hidden state and the current input and decides which information is important by outputting a value between 0 and 1 through the sigmoid function, then creates a new vector of candidate values through the tanh function, and finally multiplies the output of the sigmoid with the output of tanh to decide which information is retained and added to the cell state.

The output gate determine the value of the next hidden state. It receives information from the previous hidden state and the current input and decides which information should be output by outputting a value between 0 and 1 through the sigmoid function.

**2.2.3 Predicting the periodic term of displacement.** The periodic term in mine slope displacements refers to periodic variations in displacement driven by external environmental factors such as rainfall, groundwater infiltration pressure, and groundwater level fluctuations. These factors interact in complex ways, influencing slope stability across different time scales through changes in stress distribution, pore pressure dynamics, and material strength.

Rainfall is a primary driver of periodic displacement in mine slopes, acting through multiple interconnected mechanisms. When rainfall infiltrates the soil or rock mass, it alters the internal moisture content, increasing the self-weight of the slope while simultaneously reducing the shear strength of the soil or weathered rock. The infiltration process also leads to a rise in pore water pressure, which decreases the effective stress within the slope and weakens its structural integrity. During intense or prolonged rainfall events, the rapid accumulation of pore water pressure can temporarily accelerate slope displacement. However, after rainfall ceases, drainage processes gradually reduce the excess pore pressure, leading to a temporary stabilization or even a reversal in displacement trends.

Groundwater level fluctuations further contribute to the periodic nature of slope displacements, often exhibiting seasonal or long-term trends. As the groundwater table rises, the increased hydrostatic pressure within rock fractures and pore spaces generates additional seepage forces, exerting outward pressure on the slope. This effect can induce progressive deformation, particularly in rock masses with pre-existing discontinuities or weak structural planes. Conversely,

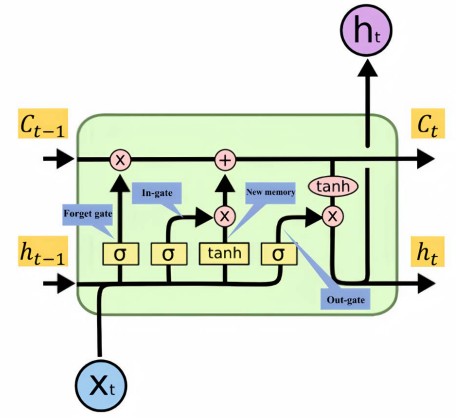

**Fig 6. Schematic diagram of the LSTM structure.**

when the groundwater level declines, the reduction in hydraulic pressure may relieve some of the stress within the slope, leading to a decrease or stabilization in displacement rates. However, in cases where desaturation causes a loss of matric suction in unsaturated soil layers, the slope may experience delayed or residual deformation, further complicating displacement patterns.

Groundwater infiltration pressure plays a critical role in slope displacement, particularly in fractured rock masses where water movement is highly anisotropic. Increased infiltration pressure due to rising groundwater levels or prolonged rainfall events can exacerbate shear stress along weak planes, promoting progressive failure. Additionally, variations in permeability between different geological layers may lead to differential pressure buildup, further influencing deformation patterns. In contrast, a drop in groundwater infiltration pressure due to seasonal drying or dewatering activities can reduce the driving forces acting on the slope, potentially enhancing its stability.

Beyond hydrological influences, external factors such as seismic activity, tectonic stress changes, and mining operations also modulate periodic displacement behavior. Seismic events generate transient dynamic loads, which can induce immediate displacement shifts or trigger long-term weakening effects through stress redistribution. Similarly, tectonic activity can introduce gradual deformation trends, interacting with hydrological processes to create complex displacement periods. Mining-induced stress redistribution, particularly in open-pit operations, alters the local stress field and can lead to incremental slope movement, further compounding the effects of environmental fluctuations.

The interaction of these factors underscores the complexity of periodic slope displacement mechanisms. Rather than acting in isolation, rainfall, groundwater dynamics, and external geophysical forces collectively shape displacement trends through interdependent processes. A deeper understanding of these interactions is essential for refining predictive models and improving the reliability of landslide early hazard warning systems.

To obtain a clear understanding of the influence of each factor on slope slippage, the link between the influencing factors and slope slippage was quantitatively analyzed by using the gray correlation method in this study. Gray relation analysis (GRA) is a multifactor statistical analysis method. The basic concept is to evaluate the closeness of connections based on the similarity of the geometric shapes of the sequence curves. The more similar the curves are, the stronger the correlation between the corresponding sequences. Conversely, less similarity indicates weaker correlation. This method is suitable for qualitative and quantitative analysis of the interrelationship between various factors in the system to provide a scientific basis for decision-makers and optimize the performance of the system. The formula for gray correlation analysis is as follows:

$$\xi_i(k) = \frac{\min_i \min_k \Delta_i(k) + \rho \max_i \max_k \Delta_i(k)}{\Delta_i(k) + \rho \max_i \max_k \Delta_i(k)}$$

(4)

where $\xi_i(k)$ is the resolution factor, $x_i$ is the ith column of data, k is the kth value in the ith column, $\rho$ is the resolution factor, with $0 \leq \rho \leq 1$, generally taken as 0.5, and $\Delta_i(k) = |y(k) - x_i(k)|$.

Understanding and analyzing the pattern of change in these periodic terms of displacement is highly important for assessing slope stability, predicting displacement trends, and developing corresponding monitoring indicators. Typically, the deformation characteristics of mine slopes increase suddenly during periods of heavy rainfall and falling groundwater levels, and the rate of landslide displacement decreases when these influences disappear. Therefore, the fluctuations in the groundwater level and osmotic pressure in the current month relative to those in the previous month are considered input factors. Considering the lag in the response of landslide deformation to rainfall, the rainfall of the current month and the previous month are considered input factors.

The SARIMA is a time series forecasting method that is an extension of the Autoregressive Integrated Moving Average (ARIMA) model and is specifically designed to address time series data with seasonal patterns. The ARIMA model captures the trend and seasonality of the time series data by considering the autoregressive (AR) term, difference (I), and

moving average (MA) term to capture the trend and seasonality of the time series data. The SARIMA model adds seasonal parameters to this model to increase its adaptability to seasonal data. The details of the parameters are shown in Table 1.

The seasonal component of the SARIMA model makes it particularly suitable for forecasting periodic terms. By considering seasonal patterns in time series data, the SARIMA model can better predict periodic fluctuations and trends, making its performance superior when dealing with data that are periodic in nature. This property makes the SARIMA model widely used for forecasting in economy, meteorology, and sales. The working principle of the model is shown in Fig 7.

**2.2.4 LSTM-SARIMA hybrid model.** The LSTM-SARIMA hybrid model demonstrates significant advantages by combining the powerful nonlinear modeling capability of LSTM with the excellent handling of seasonality and trends provided by SARIMA. LSTM excels in capturing long-term dependencies and nonlinear relationships within time series data, dynamically adjusting the state of its memory units to capture complex patterns and abrupt changes in slope displacement. This enables it to account for nonlinear impacts caused by sudden environmental changes or unexpected events, such as blasting vibrations, earthquakes, or shifts in mining activities, which can lead to sharp variations in

**Table 1. Details of the SARIMA parameters.**

| Parameters | Conceptual | Specificities |
|---|---|---|
| p | Nonseasonal autoregressive order | Indicates the correlation between observations in a time series and their past values |
| q | Nonseasonal moving average order | Nonseasonal moving average order。 |
| d | Differential order | Denotes the several orders of differencing the time series to make it a smooth time series. |
| P | Seasonal autoregressive (SAR) Order | Indicates the correlation between a seasonal component and its past seasonal components. |
| Q | Seasonal moving average (SMA) order | Denotes the correlation between the seasonal component and the error of its past seasonal component. |
| D | Seasonal difference (SI) order | Denotes the several orders of differencing the seasonal component to make it a smooth time series. |
| S | Seasonal periodic length | The period size of this seasonal series |

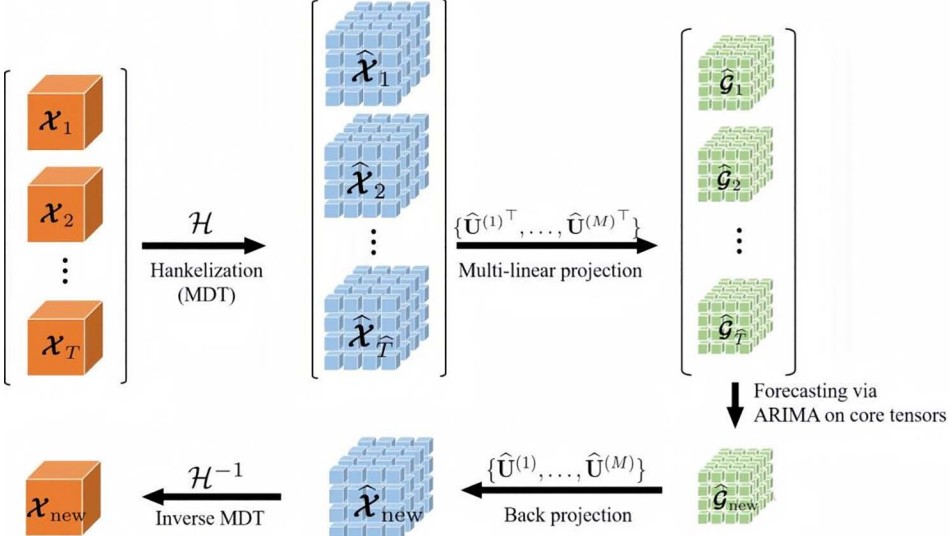

**Fig 7. Schematic diagram of the SARIMA algorithm flow.**

displacement patterns that traditional linear models often fail to capture. However, through its adaptive learning ability, LSTM can automatically learn these complex patterns from historical data, enhancing the model's accuracy and flexibility. Although LSTM is inherently a black-box model, visualizing its internal states (such as weights and activation values) and performing local sensitivity analysis can help identify which input features (such as lithology, weathering degree, etc.) play a significant role in predicting displacement.

On the other hand, SARIMA effectively captures linear trends and seasonal characteristics within time series, providing a stable baseline prediction for the model. SARIMA is capable of handling long-term periodic changes, such as precipitation periodic and temperature fluctuations, and can also identify linear trends, such as the gradual changes in slope stability over time. This feature allows SARIMA to offer a robust foundation for the model by explicitly accounting for seasonal and trend components that explain long-term variations in the time series. Since SARIMA relies on linear relationships, its coefficients are directly interpretable. The seasonal and trend components of the model can help explain periodic variations and long-term stability trends in slope displacement.

By combining the nonlinear modeling capabilities of LSTM with the linear modeling strengths of SARIMA, the hybrid model can simultaneously handle both short-term sudden events and long-term seasonal changes when facing complex slope displacement variations. This results in a more comprehensive description of the dynamic changes in slope displacement. This integration makes the LSTM-SARIMA hybrid model not only highly accurate in predicting slope displacement but also provides more intuitive explanatory support for engineering practice.

**2.2.5 Evaluation criteria for prediction algorithms.** Scholars usually use the signal-to-noise ratio (SNR) to evaluate the performance of data noise reduction models. The SNR is the ratio of the intensity of the received useful signal to the intensity of the received interfering signals (noise and interference), typically expressed in decibels (dB). A higher SNR indicates better signal quality and more accurate, clear information. Generally, an SNR of 10 dB or more is sufficient for normal operation, while an SNR of 30 dB or more ensures reliable data transmission. The formula for calculating the SNR is as follows:

$$SNR = 10 \lg \frac{P_s}{P_n} \tag{5}$$

Where Ps is the power of the signal and Pn is the power of the noise.

For the evaluation of the performance of intelligent prediction algorithms, this paper uses $R^2$ as an evaluation criterion. $R^2$ is often referred to as the coefficient of determination, and it quantifies the variance of one independent variable with respect to another. $R^2$ is the square of the Pearson's correlation coefficient, r, and it measures the linear correlation between 2 variables, X and y [25]. The expression for $R^2$ is as follows:

$$R^2 = 1 - \frac{\sum (y_i - y_{reg})^2}{\sum (y_i - \bar{y})^2} \tag{6}$$

where $y_i$ is the value of each data point, $\bar{y}$ is the average value; and $y_{reg}$ is the value predicted by the regression model.

In this paper, the results are also quantified using the mean absolute error (MAE) and root mean square error (RMSE) with the following expressions:

$$E_{MA} = \frac{1}{n} \sum_{i=1}^{n} |y_i - \bar{y}| \tag{7}$$

$$E_{RMS} = \sqrt{\frac{1}{n} \sum_{i=1}^{n} \left( y_{test}^{(i)} - \hat{y}_{test}^{(i)} \right)^2} \tag{8}$$

where $y_{test}^{(i)}$ is the output of the ith sample on the test set and $\hat{y}_{test}^{(i)}$ is the average of the ith sample on the test set. The mean absolute error $E_{MA}$ measures the average of all absolute errors, and the root mean square error $E_{RMS}$ characterizes the deviation of the prediction results.

## 2.3 Early hazard warning indicators of slope slippage

The Saito three-stage model is an empirical approach to landslide prediction, based on the analysis of landslide deformation rates over time. This model categorizes the deformation process into three distinct stages, each characterized by a specific deformation rate: the deceleration deformation stage, the uniform deformation stage, and the acceleration deformation stage. The acceleration deformation stage is further subdivided into the initial acceleration stage, the uniform acceleration stage, and the pro-slip stage, as illustrated in Fig 8. The primary purpose of this model is to describe the acceleration phase of landslide deformation prior to destabilization, enabling early hazard warning and prediction of landslide occurrences.

Numerous cases indicate that when the tangential angle of the displacement-time curve approaches 90°, slope failure begins, subsequently leading to regional landslides. The tangential angle at which failure typically occurs is generally between 89° and 89.5°. However, in practice, scholars have observed that the horizontal and vertical coordinates of the displacement-time curve can vary due to differences in scale. When one of the coordinates is stretched or compressed because of changes in scale, both the shape of the three-stage deformation curve and the tangential angle of the curve will be altered. As a result, relying solely on the tangential angle of the displacement-time curve to determine the timing of slope failure is both unreasonable and inaccurate [26].

In response to the problem of different dimensions which leads to the lack of accuracy in landslide early hazard warning and forecasting, a new idea is proposed based on the displacement-time curve.This method uses coordinate transformation to standardize the dimensions of both coordinates, ensuring the correct determination of the tangential angle. This standardization allows for the establishment of a more precise criterion for landslide forecasting.

Due to the property that the deformation rate is a stable value in the uniform deformation stage, the dimensional transformation of the horizontal and vertical coordinates is realized by using the constant rate as an intermediate medium. To analyze the homogeneous deformation stage, the cumulative displacement S in this stage is linearly distributed with the time t, i.e., s = vt, and v is the deformation rate in this stage, which is a constant value. Therefore the dimensional reset of

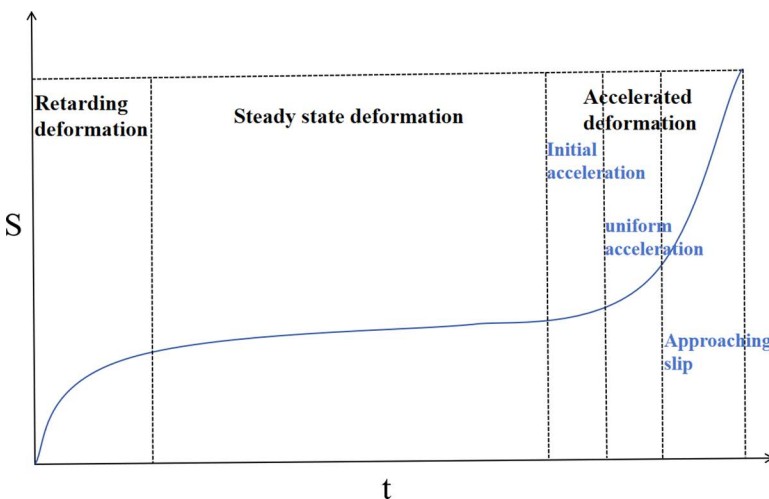

**Fig 8. Three stages of peristaltic deformation.**

the y-axis displacement is realized through the constant rate to obtain the same dimension as the x-axis time coordinate, which can be obtained:

$$T = \frac{s}{v}$$

(9)

where s is the cumulative displacement in a certain time period; v is the average velocity of the homogeneous deformation stage; T is the relative time corresponding to the true monitoring time.

The curve of relative time T relative to the true monitoring time t can be obtained by the transformation of Eq. (9), as shown in Fig 9. According to the T-t curve obtained after the dimensional transformation, the improved expression for the tangential angle $\alpha_i$ is then obtained:

$$\alpha_i = \arctan \frac{T(i) - T(i-1)}{t_i - t_{i-1}}$$

(10)

where $\alpha_i$ is the improved tangential angle obtained from the T-t curve; T(i) is the relative time; and ti is the true monitoring time.

Each stage of the T-t curve in Fig 9 corresponds to Fig 8, AB, decelerated deformation stage; BC, uniform deformation stage; CF, accelerated deformation stage; CD, initial acceleration stage; DE, uniform acceleration stage; EF, critical slip stage. The proposed landslide warning guidelines based on the tangential angle are shown in Table 2.

## 3. Result

### 3.1 Project profile

This paper presents 2 years of data monitoring and analysis work on a large open-pit mine in eastern China. The mine is located in the ancient uplift of the Mesozoic terrestrial volcanic basin margin. The main fracture systems in the mine area are three groups—northeast oriented, northwest oriented and nearly east–west oriented—all of which are manifested as

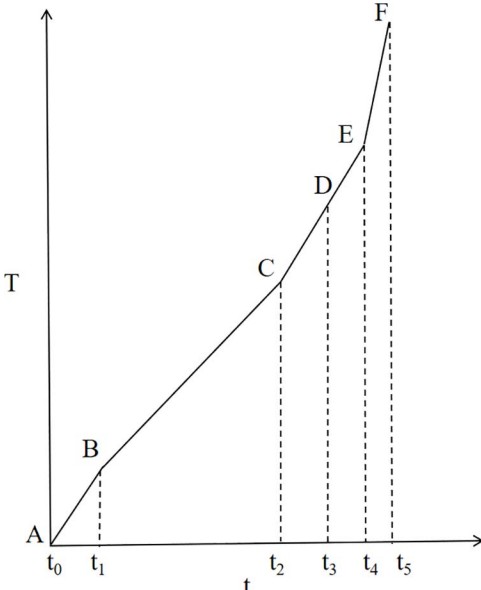

**Fig 9. T–t curve.**

**Table 2. Improve the range of definition of tangent angle.**

| $\alpha_i$ | slope condition |
|---|---|
| $\alpha_i < 45°$ | the slope deformation is in the initial deformation stage |
| $\alpha_i \approx 45°$ | the slope deformation is in the uniform deformation stage |
| $\alpha_i > 45°$ | the slope is undergoing accelerated deformation |
| $\alpha_i > 80°$ | the slope deformation enters the uniformly accelerated deformation stage |
| $\alpha_i > 85°$ | the slope deformation is in the critical slip stage, and the warning is necessary |
| $\alpha_i \approx 89°$ | the slope begins to slide |

the late stage of mineralization and play a destructive role in the ore body. Northeast-oriented fractures, which are characterized by compression and torsion, are the main fractures in mining areas and are characterized by long-term inheritance and development. The intersection of early northeast-oriented fractures and northwest-oriented fractures not only controls the formation of volcanic bodies but also provides tectonic space for the intrusion of subdiagonal granite porphyry rocks. The length of the strike is generally 500~2000 m, and the dip angle is generally 65~80°. There are F1, F2, F3, F4, F5, and F6 faults in the northeast direction.

The rock slope under investigation is located on the southern side of the mining area and is nearly linear, with an average width of approximately 670m. According to the profile characteristics, the highest elevation is +121.50m, the lowest elevation is -350.00m, and the maximum height difference is 471.50m. The overall slope angle is approximately 41.32°. The lithology primarily consists of mixed gneiss and mixed granite. The slope is affected by a fault (F4) with a strike of SE60~70° and a dip angle of 75°. Layered joint structures are well-developed along the slope, with dip angles ranging from 30° to 60°. The rock mass structure is predominantly blocky.The field photographs of the rock slope and the corresponding profile are shown in Figs 10 and 11.

The mine utilizes an R/HYB2000 non-contact slope radar with an operational range of 5.5 km, a monitoring coverage of 110°×42°, and a detection accuracy of no less than 0.1mm. The radar is installed on the northern side of the open-pit mine, as shown in Fig 12, to monitor surface displacement changes on the southern slope. The monitoring data used in this study include slope displacement data obtained from radar and various sensors, as well as rainfall, groundwater level, and groundwater osmotic pressure data. Data were collected at a time interval of 30 minutes, resulting in a total of 36,000 monitoring datasets.

### 3.2 Displacement data preprocessing

First, the radar displacement monitoring data of the open pit mine in the past two years were collected and organized, and the displacement–time curves were plotted, as shown in Fig 13. The data preprocessing work was carried out on the

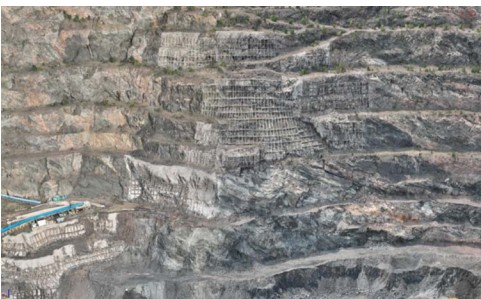

**Fig 10. Rock slope site photos.**

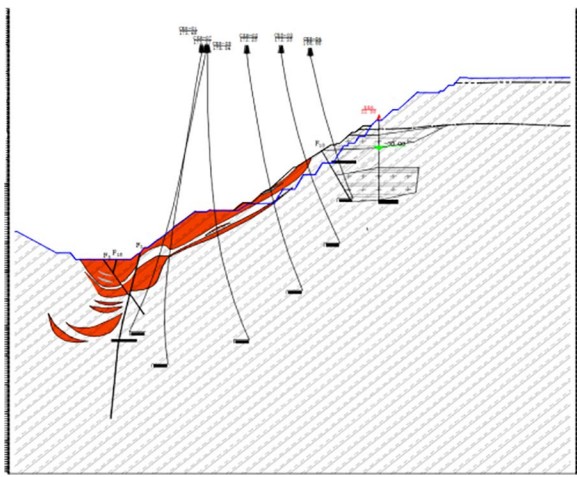

**Fig 11. Engineering geological profile.**

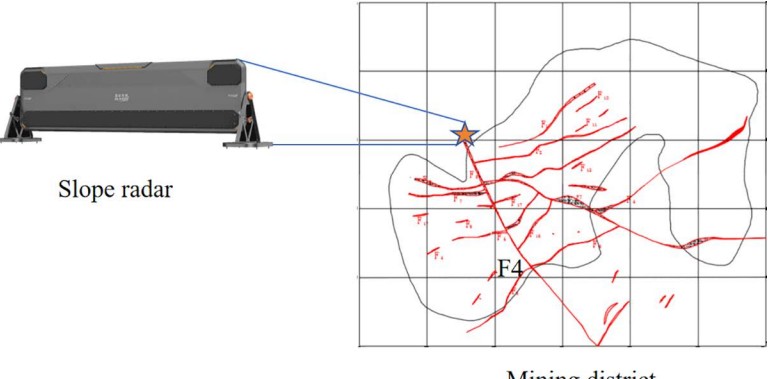

Slope radar

F4

Mining district

**Fig 12. Schematic diagram of the distribution of major fracture structures in the mine area and the location of the radar arrangement.**

monitoring data, noise reduction was performed on the original data using the sliding average filtering–wavelet denoising method, and the results of the noise reduction are shown in Figs 14 and 15.

Fig 14 shows that the original data after the first round of sliding average filtering for denoising have a better noise reduction effect for the global data, but the noise reduction effect for the local area is poor. Between July 2021 and July 2022, the data are noisy, and the sliding average filtering algorithm has a poorer noise reduction effect on the data during this time period. Therefore, the wavelet denoising method is used to perform secondary noise reduction on the noise-reduced data, and the wavelet noise reduction algorithm can focus on the local high-frequency signal region. After two rounds of noise reduction, the signal-to-noise ratios after sliding average filter noise reduction and wavelet denoising are 36 and 44, respectively, and the data noise reduction effect is good.

### 3.3 Hybrid data-driven model predictions

Trend term–periodic term separation was performed on the noise-reduced data, and analysis was performed using a hybrid data-driven model for the respective characteristics of the trend term of displacements and the periodic term of

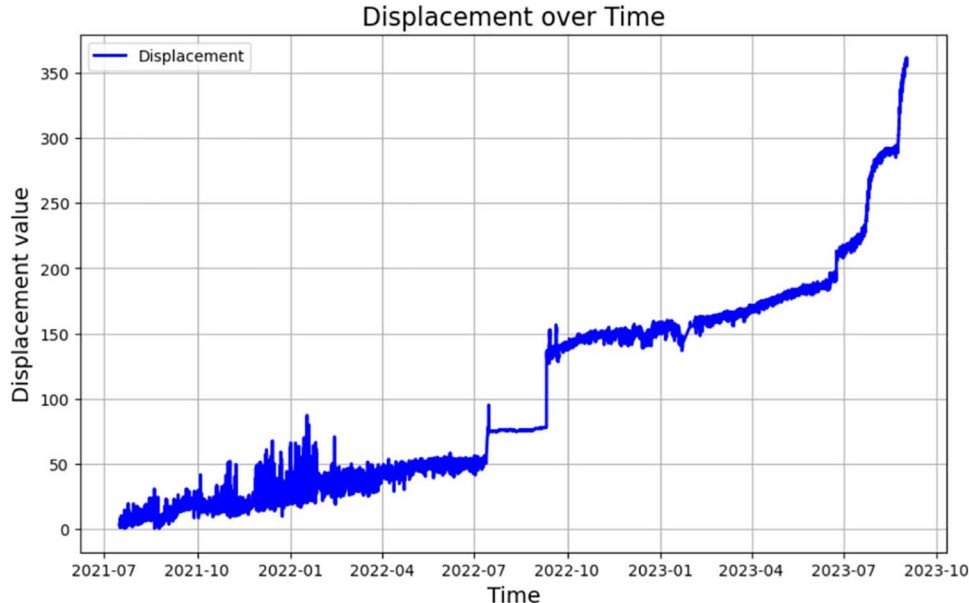

**Fig 13. Radar displacement monitoring data.**

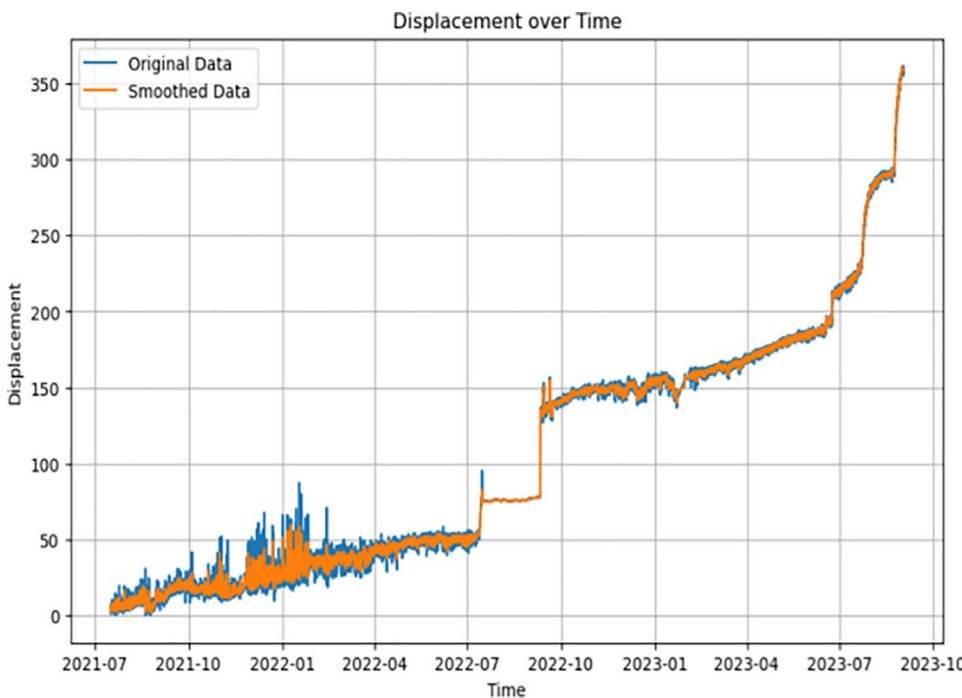

**Fig 14. Sliding average filtering global denoising.**

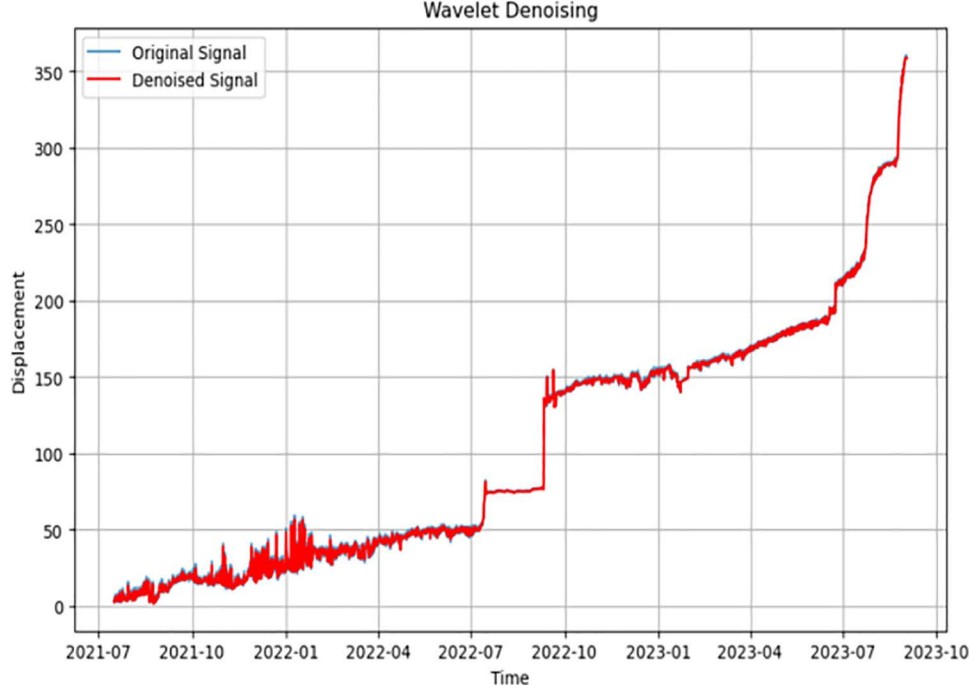

**Fig 15. Wavelet localized region focused denoising.**

displacements. Trend term–periodic term separation was performed on 2 years of monitoring data using the HP filter, and the results are shown in Fig 16.

After decomposing the displacement data into trend and periodic components, a total of 36,000 data groups were obtained for each type of displacement. Both datasets were split in an 8:2 ratio, with 80% allocated for training and 20% for testing. A hybrid data-driven model was then applied to analyze the displacement characteristics.First, the trend term of displacement was analyzed, and the LSTM time series algorithm was used to learn the changes in the trend term of displacement of the mine for the previous two years to analyze the development trend of slope displacements under the influence of the self-weight condition and to use the learned displacement trend to predict the displacement in the future time step. The results of the trend term of displacements are shown in Fig 17.

Second, a time series analysis of the period term of displacement is needed. The period term of displacement is affected by rainfall, groundwater and other factors. First, based on the collected slope displacement data of an open pit mine, groundwater fluctuation data, and rainfall and groundwater osmotic pressure data, the associations between various influencing factors and slope slippage were quantitatively analyzed using the gray correlation method. The correlation between each influencing factor and the slope slip is shown in Fig 18.

The improved SARIMA model was used to establish a time series analysis model of the periodic term of displacement using the fluctuations in groundwater level and osmotic pressure in the current month relative to the previous month and the rainfall in the current and previous months as inputs and the future periodic term of displacement as outputs over the past two years at this mine; the prediction results are shown in Fig 19.

With every half hour as a time step, the LSTM–SARIMA hybrid data-driven model was used to predict the trend term of displacement and the period term of displacement for the next 20 time steps, and the prediction results were summed to obtain the total slope displacement. To determine the excellence of the hybrid model, this paper uses the SVM, Random Forest, XGBoost, RNN and LightGBM models to predict the slope displacements for comparison. Figs 20–25 shows an

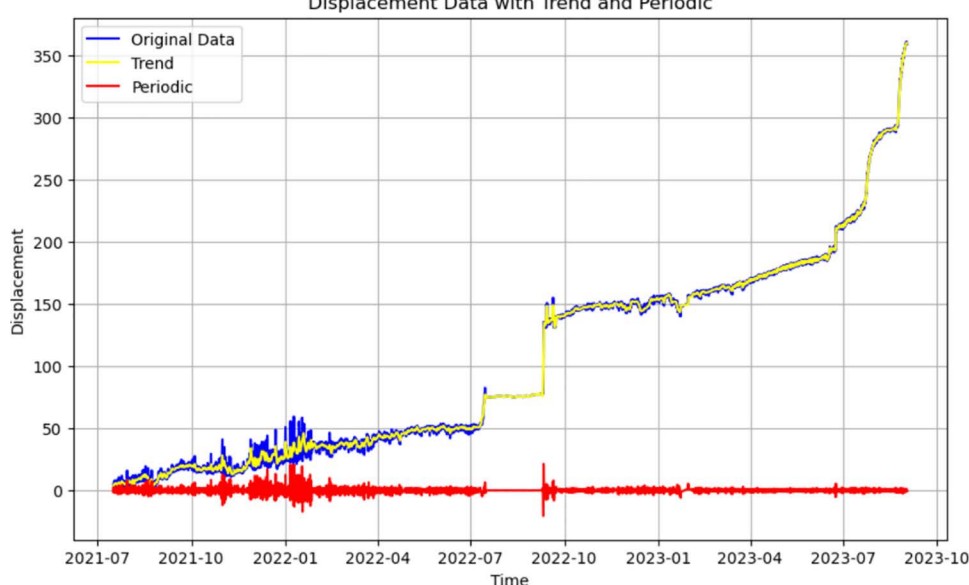

**Fig 16. Results of separating the trend and periodic terms of displacement.**

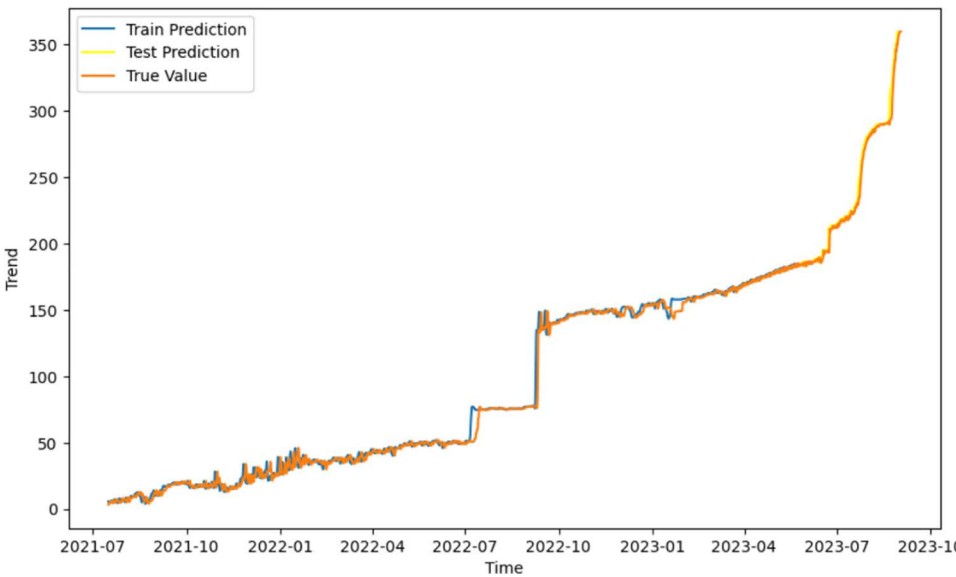

**Fig 17. Future time step prediction results of the trend term of displacement.**

error line plot of the displacement data obtained from the four data-driven model predictions plotted against the observed data on the day before the landslide.

As shown in Figs 20–25, according to the length of the error line at each prediction point of the error line graph, the hybrid LSTM–SARIMA data-driven model has the smallest error among the four models. The performance results of the four data-driven models are shown in Table 3.

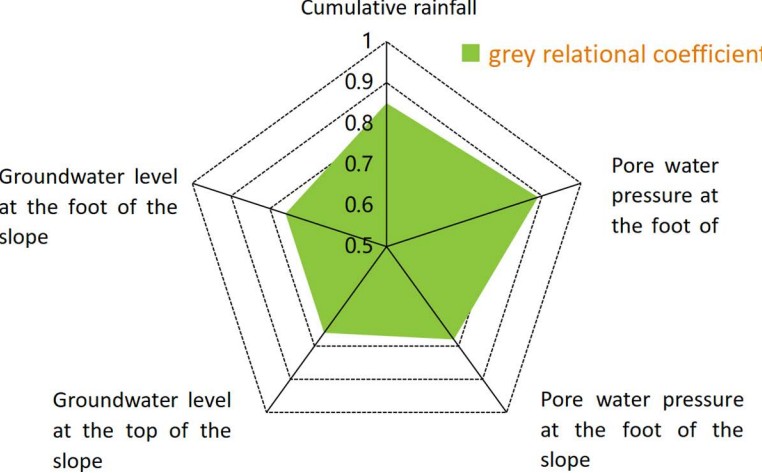

**Fig 18. Gray correlation coefficient calculation results.**

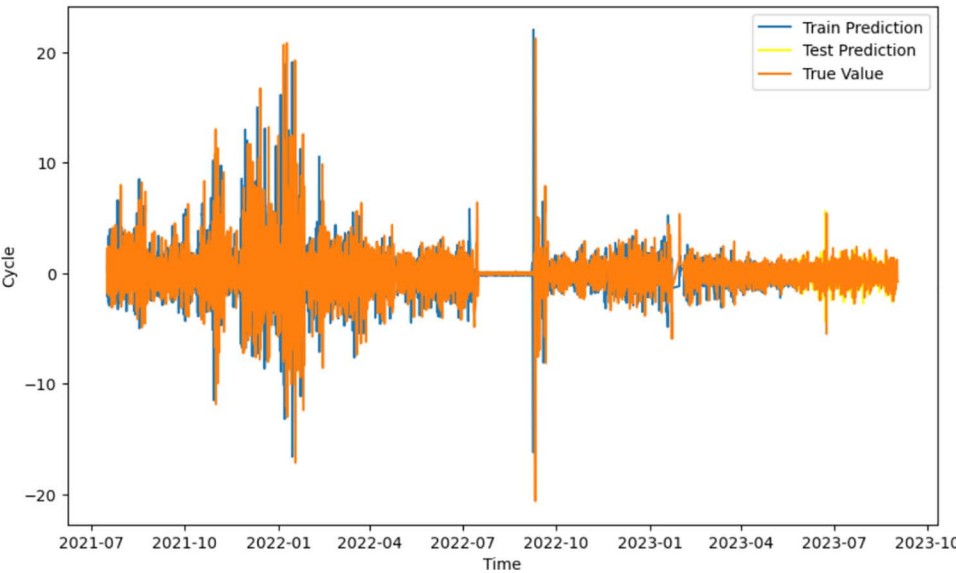

**Fig 19. Prediction results of the future time step of the periodic term of displacement.**

### 3.4 Early hazard warning indicators

After completing the slope displacement prediction, the potential slip is determined. The obtained S–t displacement curve is improved and transformed to obtain the T–t curve, which is shown in Fig 26.

First, after obtaining the T–t curve, it is necessary to divide the whole curve into stages. The deformation rate of the AB section fluctuates, the deformation rate of the whole stage shows a rapid growth trend, and the deformation rate decreases at the end of the time period and tends to level off; thus, the AB section is defined as the initial deformation stage. The deformation rate of the BC section is flat, and the tangential angle is approximately 45°, which is defined as the uniform deformation stage. In the DF section, the overall deformation rate rises significantly, with the tangential angle exceeding 45°. This phase is identified as the accelerated deformation stage. In the whole process of deformation stage

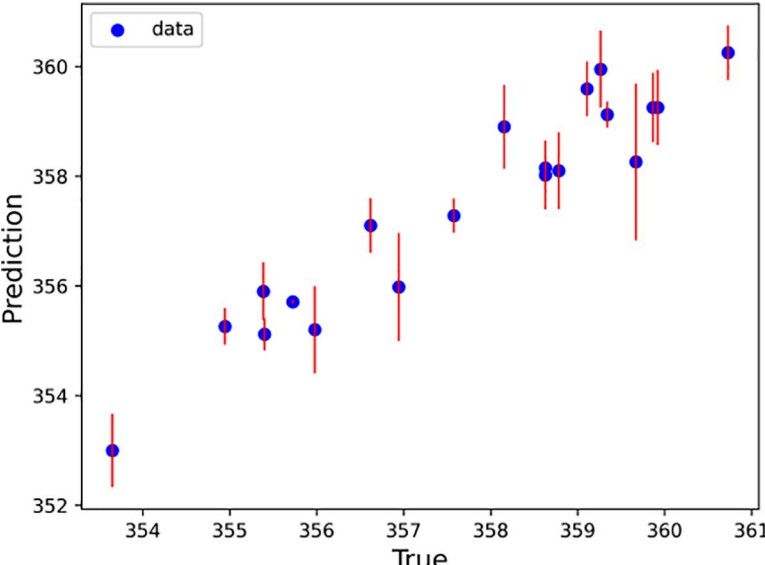

**Fig 20. Error line diagrams for LSTM–SARIMA hybrid models.**

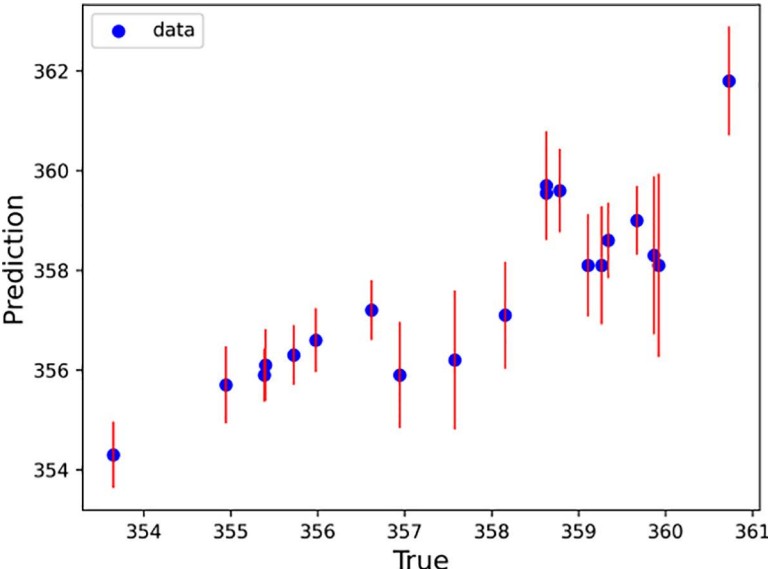

**Fig 21. Error line diagrams for RNN.**

delineation, the most important thing is to determine the accelerated deformation stage and to analyze it in a focused manner.

In Fig 26, the T-t curve shows the complete deformation process of the slope from deceleration deformation to final destruction. However, in the actual project, due to the different time of installation of the monitoring equipment, the time of installation may be located in the early stage of deformation, or the uniform deformation stage or even the destruction stage.In addition, the decelerated deformation phase of a slope may last for a long or short period of time.As a result, the

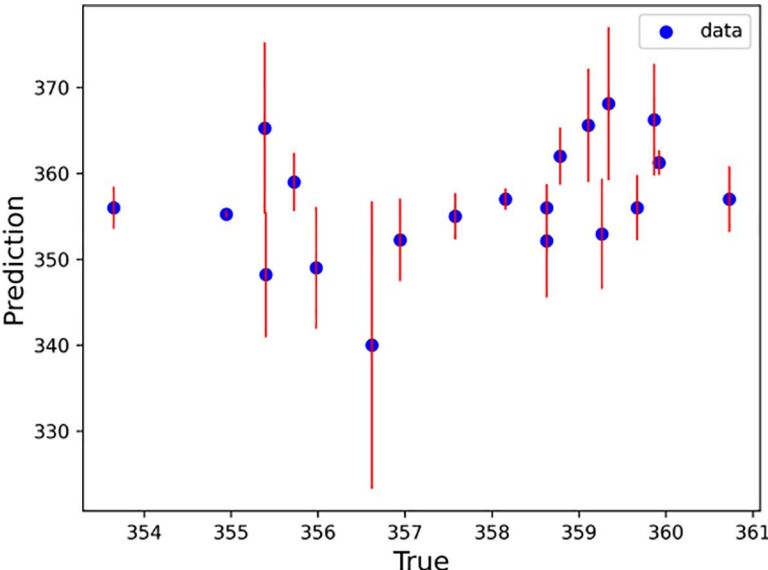

**Fig 22. Error line diagrams for SVM.**

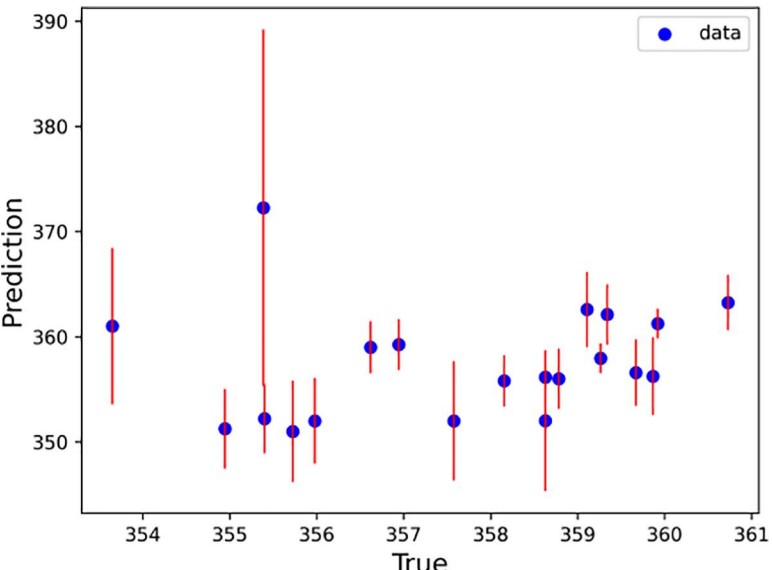

**Fig 23. Error line diagrams for XGBoost.**

displacement-time curves of many actual slopes might only display the phases of uniform and accelerated deformation, omitting the initial decelerated phase. Sometimes only curves for the accelerated deformation phase are obtained.

According to Table 2.When slope deformation enters the accelerated deformation stage, the improved tangential angle is obviously greater than 45°. When the improved tangential angle exceeds 80°, the slope deformation rate increases obviously. When the improved tangential angle exceeds 85°, the deformation rate and the improved tangential angle tend

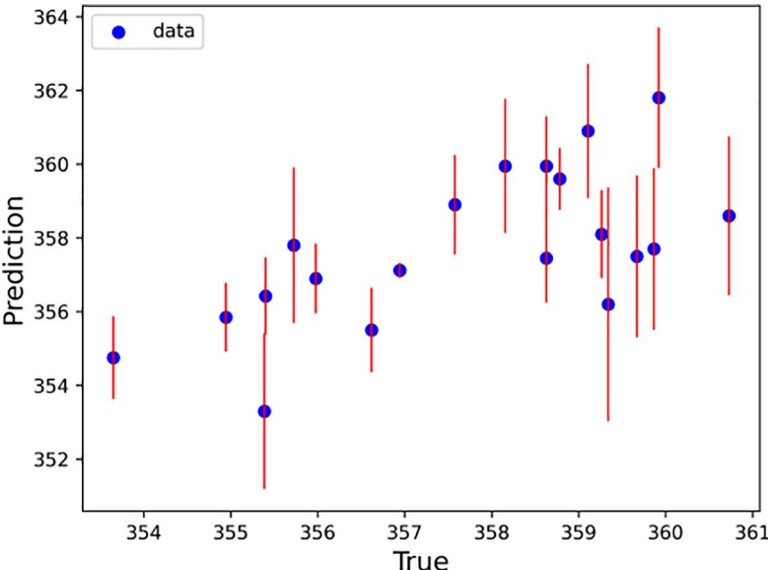

**Fig 24. Error line diagrams for LightGBM.**

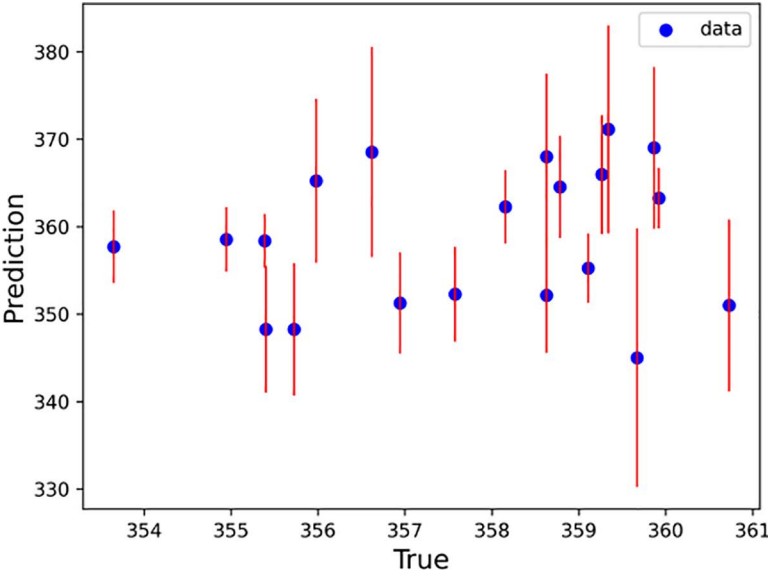

**Fig 25. Error line diagrams for Random forest.**

to increase sharply with time, and the slope shows obvious signs of critical damage, which requires warning. When the tangential angle reaches approximately 89°, the slope begins to slide.

Therefore, based on Fig 26, a relatively quantitative criterion for the division of the three substages of the accelerated deformation stage can be formulated using the improved tangential angle. When the improved tangential angle is greater than 45°, slope deformation of this open pit mine enters the primary accelerated stage, known as the CD segment.When the enhanced tangential angle exceeds 80°, the slope deformation transitions into the uniformly accelerated deformation

**Table 3. Results of the four data-driven models.**

| Model | R² | RMSE | MAE |
|---|---|---|---|
| LSTM–SARIMA hybrid models | 0.96 | 0.785 | 1.642 |
| RNN | 0.86 | 1.152 | 2.086 |
| SVM | 0.84 | 1.263 | 2.154 |
| XGBoost | 0.81 | 1.496 | 2.225 |
| LightGBM | 0.77 | 2.129 | 2.954 |
| RF | 0.72 | 2.342 | 3.113 |

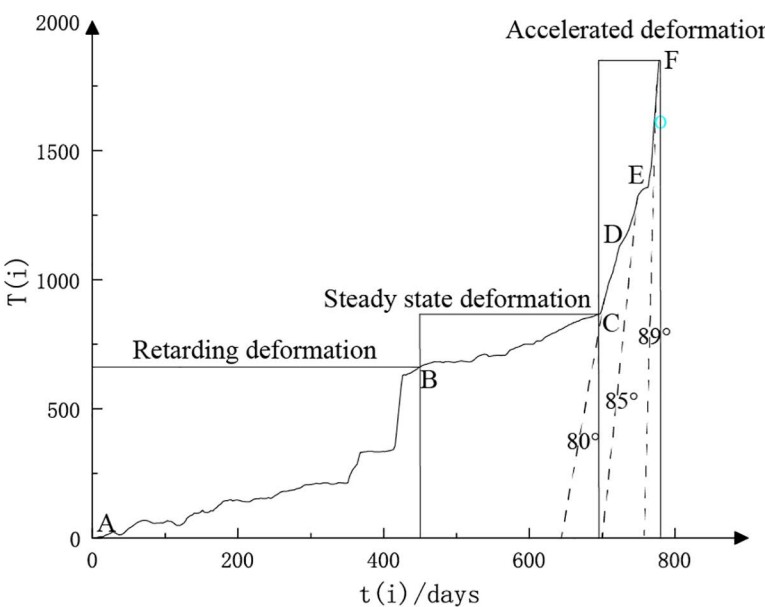

**Fig 26. T–t curve for open pit displacement monitoring.**

stage, known as the DE segment. If the improved tangential angle surpasses 85°, the slope deformation moves into the critical damage stage, known as the EF segment. When the tangential angle reaches approximately 89°, the slope starts to experience a landslide.

Using the improved tangential angle judgment basis, this paper predicted 8 hours in advance that a landslide was about to occur on the slope at approximately 14:00 p.m. on September 16, 2023, and then the on-site staff were notified to evacuate in an orderly manner. Eventually, the landslide occurred at 15:12 p.m. on September 16, so successfully predicting the landslide disaster in the open-pit mine was successfully predicted. The landslide site is shown in Fig 27.

Landslide early hazard warning is a complex, interlinked process where the absence or improper application of any component can lead to inaccurate warnings. This paper presents a comprehensive technical framework for landslide disaster early hazard warning in rock slope, addressing the entire early hazard warning process rather than isolated steps. The proposed framework integrates data analysis, landslide mechanism investigation, intelligent predictive model development, and early hazard warning indicator establishment, drawing on multidisciplinary expertise to create a complete landslide early hazard warning system. This system has been validated in real-world projects, demonstrating its accuracy and potential for application across multiple mining sites to enhance mine safety and support secure production.

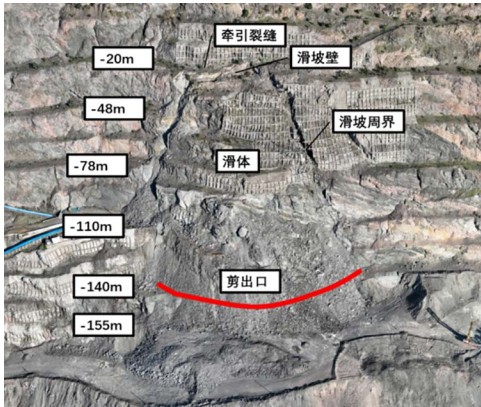

**Fig 27. Landslide site and landslide boundary diagram of open pit research area.**

## 4. Discussion

Although the proposed landslide early hazard warning framework has demonstrated significant improvements in prediction accuracy and practical applicability, several limitations remain when compared to previous studies. One of the main challenges is the dependency on high-quality monitoring data. While noise reduction techniques, such as sliding average and wavelet denoising, enhance data reliability, the effectiveness of the warning system can still be affected by missing or inconsistent data, particularly in extreme weather conditions or complex geological settings.

Furthermore, the hybrid AI-based model integrates physical insights with machine learning algorithms, improving prediction accuracy. However, the interpretability of AI-driven models remains a challenge. Unlike purely physics-based approaches, machine learning models often function as black boxes, making it difficult to directly correlate predicted outcomes with underlying geomechanical processes. Future research should focus on enhancing the explainability of AI models by incorporating physical constraints and causal relationships within the learning framework.

This study provides a systematic and practical landslide early hazard warning approach that combines monitoring, modeling, and intelligent prediction. Addressing the existing limitations will further enhance the robustness and applicability of this framework, contributing to more reliable and standardized early hazard warning systems in open-pit mining and other geotechnical settings.

## 5. Conclusion

In this paper, a hybrid data-driven model is developed to predict future spatial and temporal trends of slope displacement through the collection, processing, and analysis of slope displacement monitoring data. An improved tangent angle is introduced as an early hazard warning indicator, ultimately creating a comprehensive technical system for precise early hazard warning of potential landslide hazards in rock slope. Based on the research in this paper, the following conclusions are drawn:

1. The single-data noise reduction model has a poor noise reduction effect on the data as a whole. This paper proposes a hybrid sliding average–wavelet noise reduction model to focus on the global and local strong noise regions, and the noise reduction effect is good. The signal-to-noise ratios after noise reduction by the hybrid noise reduction model are 36 and 44.

2. The LSTM–SARIMA hybrid data-driven model is used to predict the trend term and period term of slope displacement separately, and the prediction accuracy is as high as 96%, which is even better than that of the single data-driven

model. The untimely warning and casualties caused by the inaccurate prediction of future time step displacement data can be effectively solved.

3. Using the improved tangential angle of the T–t curve as a warning indicator addresses the inaccuracies that arise when the traditional tangent angle of the S–t curve is used, due to potential differences in the units and scales of the displacement-time curve coordinates. When the improved T–t tangential angle surpasses 85°, the slope exhibits clear signs of impending failure, indicating the need for a warning. As the T–t tangential angle approaches 89°, the slope starts to slide.

## Supporting information

**S1 Data.**
(XLSX)

**S2 File.**
(PY)

**S3 File.**
(IPYNB)

**S4 Data.**
(XLSX)

**S5 Data.**
(XLSX)

**S6 Data.**
(XLSX)

**S7 Data.**
(XLSX)

**S7 Data.**
(XLSX)

**S8 Data.**
(XLSX)

**S9 Data.**
(XLSX)

**S10 File.**
(IPYNB)

**S11 Data.**
(XLSX)

**S12 Data.**
(XLSX)

**S13 Data.**
(XLSX)

## Author contributions

**Data curation:** Yongxin Dai.

**Methodology:** Zijian Li.

**Project administration:** Jingbiao Lu.

**Writing – original draft:** Zijian Li.

**Writing – review & editing:** Yongxin Dai.

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
