## [Decision Letter · Decision Letter 0]

14 Jan 2025

Dear Dr. Li,

Thank you for submitting your manuscript to PLOS ONE. After careful consideration, we feel that it has merit but does not fully meet PLOS ONE’s publication criteria as it currently stands. Therefore, we invite you to submit a revised version of the manuscript that addresses the points raised during the review process.

**ACADEMIC EDITOR:**

The review process is now complete, and we have two reports submitted by the expert reviewers. As can be seen from the reports, both reviewers request major revisions before it is reviewed again. In particular, reviewers #1 and #2  proposed many critical comments on applying technical terms, research summary, research innovations, model comparisons, and research depths in this study. Thus,  I invite you to revise the submitted manuscript carefully.

Meanwhile, I want to thank the reviewers for their great efforts on your manuscript and you for your submission.

We look forward to receiving your revised manuscript.

Kind regards,

Linwei Li

Academic Editor

PLOS ONE

Journal Requirements:

2. Please note that PLOS ONE has specific guidelines on code sharing for submissions in which author-generated code underpins the findings in the manuscript. In these cases, we expect all author-generated code to be made available without restrictions upon publication of the work. Please review our guidelines at https://journals.plos.org/plosone/s/materials-and-software-sharing#loc-sharing-code and ensure that your code is shared in a way that follows best practice and facilitates reproducibility and reuse

4. We note that Figure 8 in your submission contain [map/satellite] images which may be copyrighted. All PLOS content is published under the Creative Commons Attribution License (CC BY 4.0), which means that the manuscript, images, and Supporting Information files will be freely available online, and any third party is permitted to access, download, copy, distribute, and use these materials in any way, even commercially, with proper attribution. For these reasons, we cannot publish previously copyrighted maps or satellite images created using proprietary data, such as Google software (Google Maps, Street View, and Earth). For more information, see our copyright guidelines: http://journals.plos.org/plosone/s/licenses-and-copyright .

   a. You may seek permission from the original copyright holder of Figure 8 to publish the content specifically under the CC BY 4.0 license. 

5. We note that you have indicated that there are restrictions to data sharing for this study. PLOS only allows data to be available upon request if there are legal or ethical restrictions on sharing data publicly. For more information on unacceptable data access restrictions, please see http://journals.plos.org/plosone/s/data-availability#loc-unacceptable-data-access-restrictions .   

Additional Editor Comments :

NaN

Reviewers' comments:

Reviewer's Responses to Questions

**Comments to the Author**

1. Is the manuscript technically sound, and do the data support the conclusions?

Reviewer #1: Yes

Reviewer #2: Yes

2. Has the statistical analysis been performed appropriately and rigorously?

Reviewer #1: No

Reviewer #2: Yes

3. Have the authors made all data underlying the findings in their manuscript fully available?

Reviewer #1: No

Reviewer #2: No

4. Is the manuscript presented in an intelligible fashion and written in standard English?

Reviewer #1: No

Reviewer #2: Yes

Reviewer #1: 1. The title of the paper 'A Novel Method... Hybrid Data Driven Model' is too vague. Suggest modifying it to a specific method statement.

2. The authors may have confused the concept and difference of the specialized term “hazard” and “risk”. From the perspective of the main content, this work only involves “hazard”. Please refer to previous publications and follow the landslide risk assessment framework. For example: van Westen et al., 2006, https://doi.org/10.1007/s10064-005-0023-0, Wang et al., 2021, https://doi.org/10.3390/rs13132625�Wen et al., DOI: 10.1016/j.jenvman.2023.118177. Suggest modifying the relevant terms throughout the context, such as: the "risk" should be "hazard", the "Early warning" should be "Early hazard warning".

3. The Introduction section of the manuscript does not clearly state the research gap, nor does it highlight the innovation and contribution of this work. Suggest adding the state-of-the-art literatures, such as: Doi: 10.1016/j.gsf.2024.101959, Doi: 10.1016/j.gr.2023.09.016, Doi: 10.1007/s11440-023-02050-9 to further clarify the research gap.

4. For displacement prediction models with periodic and trend terms, it should be further clarified how to divide the training and testing datasets.

5. The author should further clarify the details of the study case, the rock slope and the monitoring data (displacement).

6. What are the limitations of this work comparing relevant previous publications? Need to do in-depth analysis in the discussion section.

7. There is no unified for the specialized terms. As shown in the legend in the upper right corner of Figure 10, "Cycle" may be "periodic". It should review the entire context and further polish the writing English.

Reviewer #2: The manuscript presents a hybrid approach to landslide early warning with significant potential for practical application. By implementing the following suggested revisions, the paper can be strengthened methodologically and provide more comprehensive insights into slope stability prediction.

1. While the paper compares four models, consider adding more machine learning models or ensemble techniques for a more comprehensive comparison.

2. Elaborate on the limitations of the sliding average-wavelet noise reduction method and potential alternative approaches.

3. Provide a more in-depth theoretical explanation of how different environmental factors interact to influence slope displacement.

4. Include a section discussing the interpretability of the LSTM-SARIMA hybrid model, explaining how it captures complex nonlinear relationships.

5. Enhance the manuscript with more detailed visualizations, particularly showing the intermediate steps of data processing and model prediction.

6. Develop a more comprehensive section explicitly discussing the study's limitations and potential sources of error.

7. Furthermore, the work would greatly benefit from including and referencing more recent material on practical implementations of machine learning methods across several domains. It is recommended to engage in discussions on the comparison of results and the integration of these concepts into your work. You may find more information on this topic in the following articles: https://doi.org/10.1007/s11004-023-10116-3,
https://doi.org/10.1007/s11069-024-06917-2,
https://doi.org/10.1007/s12583-021-1525-9,
https://doi.org/10.1007/s12583-021-1407-1.

8. “Malery et al. (Malamud et al., 2010) Li Tianbin et al. (Chen et al., 2016)”: List only the first author's name followed by “et al.” in every citation.

9. More details of the open-pit mine slope should be provided. Picture and cross-section should be included.

10. Provide more details about the data collection process, including the specific radar monitoring equipment used, installed location, its precision, and calibration methods.

11. Fig. 16: please include the landslide boundary.

12. Conduct a thorough proofreading of the manuscript to correct any grammatical errors, improve sentence structure, and enhance overall clarity.

**Do you want your identity to be public for this peer review?** For information about this choice, including consent withdrawal, please see our Privacy Policy

Reviewer #1: No

Reviewer #2: No

---

## [Author Response · Author response to Decision Letter 1]

20 Mar 2025

To Reviewer 1:

1. The title of the paper 'A Novel Method... Hybrid Data Driven Model' is too vague. Suggest modifying it to a specific method statement.

Response

We thank the experts for their valuable comments and we apologize for the unclear presentation of the title of the paper. In this paper, we mainly predict the slope displacement by LSTM-SARIMA hybrid data-driven model, which is combined with the early warning criterion of improving the tangential angle to complete the accurate early warning of slope landslide disaster.

The comments made by the experts are very valuable, therefore, we modify the title of the paper to “Landslide Hazard Early Warning System for Rock Slopes Using a Hybrid LSTM-SARIMA Data-Driven Model .”

2.The authors may have confused the concept and difference of the specialized term “hazard” and “risk”. From the perspective of the main content, this work only involves “hazard”. Please refer to previous publications and follow the landslide risk assessment framework. For example: van Westen et al., 2006, https://doi.org/10.1007/s10064-005-0023-0, Wang et al., 2021, https://doi.org/10.3390/rs13132625�Wen et al., DOI: 10.1016/j.jenvman.2023.118177. Suggest modifying the relevant terms throughout the context, such as: the "risk" should be "hazard", the "Early warning" should be "Early hazard warning".

Response

Thanks for the Reviewers’ careful review. We are sorry for the paper's unclear presentation. In this paper, through the technical processes of data noise reduction, displacement component separation, slope displacement prediction, and early warning indicator establishment, we achieve accurate early warning of slope disasters. We are sorry that the expression of risk and hazard is not clear enough here, thanks to the valuable opinions of the experts, we have made corresponding modifications in many places in the paper to change the risk to hazard.

3.The Introduction section of the manuscript does not clearly state the research gap, nor does it highlight the innovation and contribution of this work. Suggest adding the state-of-the-art literatures, such as: Doi: 10.1016/j.gsf.2024.101959, Doi: 10.1016/j.gr.2023.09.016, Doi: 10.1007/s11440-023-02050-9 to further clarify the research gap.

Response

Thanks for the Reviewers’ careful review. We are sorry for the paper's unclear presentation. The introductory part of this paper focuses on the research done by experts and scholars on slope displacement prediction in the past time, mainly focusing on both mathematical methods and artificial intelligence algorithms. For the research results of scholars in recent years are lacking, therefore, this paper adds the results taken by scholars in this direction in recent years.

4.For displacement prediction models with periodic and trend terms, it should be further clarified how to divide the training and testing datasets.

Response

Thank you very much for this expert's valuable opinion, this opinion is of great value to improve the quality of our article. This paper lacks elaboration on dataset partitioning, therefore, we will explicitly state how the training and testing sets are partitioned in the paper.

5.The author should further clarify the details of the study case, the rock slope and the monitoring data (displacement).

Response

Thanks for the Reviewers’ careful review. We are sorry for the paper's unclear presentation. Further details of the study cases, rocky slopes and monitoring data will be elucidated in the text.

6.What are the limitations of this work comparing relevant previous publications? Need to do in-depth analysis in the discussion section.

Response

Many thanks to the experts for their valuable inputs. This comment is very helpful in improving the quality of the article. After describing the theory, methodology, and application, this article lacks discussion and comparison of research results. We will add a discussion section at the end of the article to clarify the effects and limitations of the research content.

7.There is no unified for the specialized terms. As shown in the legend in the upper right corner of Figure 10, "Cycle" may be "periodic". It should review the entire context and further polish the writing English.

Response

Thanks for the Reviewers’ careful review. We are sorry for the paper's unclear presentation. We will standardize terminology throughout the text.

To Reviewer 2:

The manuscript presents a hybrid approach to landslide early warning with significant potential for practical application. By implementing the following suggested revisions, the paper can be strengthened methodologically and provide more comprehensive insights into slope stability prediction.

1.While the paper compares four models, consider adding more machine learning models or ensemble techniques for a more comprehensive comparison.

Response

Thanks a lot to the experts for their valuable comments, this comment is extremely helpful in improving the quality of the article. In this paper, three machine learning models are used to do performance comparison with hybrid LSTM-SARIMA model. Based on the comments made by the experts, we added RNN and LightGBM, which are very widely used nowadays, for displacement prediction.

2.Elaborate on the limitations of the sliding average-wavelet noise reduction method and potential alternative approaches.

Response

Thanks for the Reviewers’ careful review. We are sorry for the paper's unclear presentation.We will detail the limitations and possible alternatives to the sliding average wavelet noise reduction method in this paper.

3.Provide a more in-depth theoretical explanation of how different environmental factors interact to influence slope displacement.

Response

Thanks for the Reviewers’ careful review. We are sorry for the paper's unclear presentation. The mechanism of the effect of rainfall, groundwater level, and groundwater infiltration pressure on slope displacement is complex but well documented. This paper is not clear enough about how different environmental factors interact to affect slope displacement, so we have rethought and reorganized our work in order to express the mechanism more clearly.

4.Include a section discussing the interpretability of the LSTM-SARIMA hybrid model, explaining how it captures complex nonlinear relationships.

Response

Thanks for the Reviewers’ careful review. We are sorry for the paper's unclear presentation. Therefore, we have added a subsection to the text dedicated to the section on the interpretability of the LSTM-SARIMA hybrid model, explaining how it captures complex nonlinear relationships.

5.Enhance the manuscript with more detailed visualizations, particularly showing the intermediate steps of data processing and model prediction.

Response

Thanks for the Reviewers’ careful review. We are sorry for the paper's unclear presentation. We fully understand the importance of presenting the intermediate steps of data processing and model prediction in detail, however, the inherent nature of machine learning models (especially deep learning architectures), whose internal feature representations are often automatically generated through multiple layers of nonlinear transformations, makes it notably challenging to visualize the intermediate processes. Unlike white-box models, the feature extraction paths of such models are difficult to decouple into interpretable discrete steps, and their prediction mechanisms rely more on distributed representations than on manually predefined logical rules.

In order to enhance the transparency of the methodology used in this paper, we provide relatively detailed descriptions of the mechanism, characteristics and effects of the models in Articles 2.2.1, 2.2.2 and 2.2.3, and make use of Fig. 3, Fig. 4 and Fig. 5 to provide an obvious demonstration of the computational process of various models.

In practical application, we utilize Fig. 9 to demonstrate the comparison of model effects before and after noise reduction. We also demonstrate the results of model prediction using Fig. 10 to Fig. 11, and plot the error line graph (Fig. 14) to show the performance of different machine learning models.

Once again, we thank the experts for their valuable comments. We are aware that the current visualization presentation is still lacking. However, the characteristics of the model prevented us from proceeding further. We kindly request the experts for their understanding.

6.Develop a more comprehensive section explicitly discussing the study's limitations and potential sources of error.

Response

Thanks for the Reviewers’ careful review. We are sorry for the paper's unclear presentation. We will explicitly discuss the limitations of the study and potential sources of error based on a comparison of existing research results and the findings of this paper.

7.Furthermore, the work would greatly benefit from including and referencing more recent material on practical implementations of machine learning methods across several domains. It is recommended to engage in discussions on the comparison of results and the integration of these concepts into your work. You may find more information on this topic in the following articles: https://doi.org/10.1007/s11004-023-10116-3,
https://doi.org/10.1007/s11069-024-06917-2,
https://doi.org/10.1007/s12583-021-1525-9,
https://doi.org/10.1007/s12583-021-1407-1.

Response

Thanks for the Reviewers’ careful review. We are sorry for the paper's unclear presentation. The introductory part of this paper focuses on the research done by experts and scholars on slope displacement prediction in the past time, mainly focusing on both mathematical methods and artificial intelligence algorithms. For the research results of scholars in recent years are lacking, therefore, this paper adds the results taken by scholars in this direction in recent years.

8.“Malery et al. (Malamud et al., 2010) Li Tianbin et al. (Chen et al., 2016)”: List only the first author's name followed by “et al.” in every citation.

Response

Thanks for the Reviewers’ careful review. We are sorry for the paper's unclear presentation.We have changed the corresponding expression in the text.

9.More details of the open-pit mine slope should be provided. Picture and cross-section should be included.

Response

Thanks for the Reviewers’ careful review. We are sorry for the paper's unclear presentation. Further details of the study cases, rocky slopes and monitoring data will be elucidated in the text.

10.Provide more details about the data collection process, including the specific radar monitoring equipment used, installed location, its precision, and calibration methods.

Response

Thanks for the Reviewers’ careful review. We are sorry for the paper's unclear presentation.We will add to the text an elaboration on data acquisition, including the specific radar monitoring equipment used, installation locations, accuracy, and calibration methods.

11.Fig. 16: please include the landslide boundary.

Response

Thanks for the Reviewers’ careful review. We are sorry for the paper's unclear presentation.We will show the landslide boundary.

12. Conduct a thorough proofreading of the manuscript to correct any grammatical errors, improve sentence structure, and enhance overall clarity.

Response

Thanks for the Reviewers’ careful review. We apologize for the grammatical errors in the article. We will be checking and revising the entire article to correct all grammatical errors, improve sentence structure, and improve overall clarity.

---

## [Decision Letter · Decision Letter 1]

13 Apr 2025

Landslide Hazard Early Warning Method for Rock Slopes Using a Hybrid LSTM-SARIMA Data-Driven Model

PONE-D-24-57504R1

Dear Dr. Li,

We’re pleased to inform you that your manuscript has been judged scientifically suitable for publication and will be formally accepted for publication once it meets all outstanding technical requirements.

Kind regards,

Linwei Li

Academic Editor

PLOS ONE

Additional Editor Comments (optional):

None

Reviewers' comments:

Reviewer's Responses to Questions

**Comments to the Author**

Reviewer #2: All comments have been addressed

2. Is the manuscript technically sound, and do the data support the conclusions?

Reviewer #2: Yes

3. Has the statistical analysis been performed appropriately and rigorously?

Reviewer #2: Yes

4. Have the authors made all data underlying the findings in their manuscript fully available?

Reviewer #2: Yes

5. Is the manuscript presented in an intelligible fashion and written in standard English?

Reviewer #2: Yes

Reviewer #2: (No Response)

**Do you want your identity to be public for this peer review?** For information about this choice, including consent withdrawal, please see our Privacy Policy

Reviewer #2: No

---

## [Editor Report · Acceptance letter]

PONE-D-24-57504R1

PLOS ONE

Dear Dr. Li,

I'm pleased to inform you that your manuscript has been deemed suitable for publication in PLOS ONE. Congratulations! Your manuscript is now being handed over to our production team.

Kind regards,

on behalf of

Dr. Linwei Li

Academic Editor

PLOS ONE